# Structure of dimeric lipoprotein lipase reveals a pore adjacent to the active site

Kathryn H. Gunn[1] & Saskia B. Neher[1]✉

Lipoprotein lipase (LPL) hydrolyzes triglycerides from circulating lipoproteins, releasing free fatty acids. Active LPL is needed to prevent hypertriglyceridemia, which is a risk factor for cardiovascular disease (CVD). Using cryogenic electron microscopy (cryoEM), we determined the structure of an active LPL dimer at 3.9 Å resolution. This structure reveals an open hydrophobic pore adjacent to the active site residues. Using modeling, we demonstrate that this pore can accommodate an acyl chain from a triglyceride. Known LPL mutations that lead to hypertriglyceridemia localize to the end of the pore and cause defective substrate hydrolysis. The pore may provide additional substrate specificity and/or allow unidirectional acyl chain release from LPL. This structure also revises previous models on how LPL dimerizes, revealing a C-terminal to C-terminal interface. We hypothesize that this active C-terminal to C-terminal conformation is adopted by LPL when associated with lipoproteins in capillaries.

Lipases are of critical importance due to their roles liberating free fatty acids (FFA) from triglycerides, enabling their transport across membranes. Lipoprotein lipase (LPL) is a secreted enzyme that is active in the capillaries, where it hydrolyzes triglycerides found in chylomicrons and very low-density lipoproteins (VLDL) to release FFA, which are transported across the capillary endothelial membrane as a source of energy[1]. In addition to acting on triglycerides, LPL has phospholipase activity[2], making it essential to regulate where and when LPL is active to prevent off-target hydrolysis events. LPL is synthesized in the adipose and oxidative tissues and is packaged into secretory vesicles using either a sphingomyelin-specific pathway with heparan sulfate proteoglycan (HSPG) syndecan-1 (SDC1) or a bulk flow secretion pathway[3]. LPL secretion is regulated in part by nutritional and metabolic cues from the environment, and some LPL is stored in vesicles until nutritional signaling induces secretion[4]. We have previously shown that in some storage vesicles, LPL adopts a filament-like structure that follows the vesicle membrane and resembles the helical oligomer of LPL we solved by cryoEM[5]. Following exocytosis, LPL is released from the SDC1 HSPGs, possibly due to interaction with albumin and HSPGs in the interstitial space[6]. LPL transits the interstitial space to reach glycosylphosphatidylinositol anchored high-density lipoprotein binding protein 1 (GPIHBP1), which is expressed by capillary endothelial cells[7]. GPIHBP1 is a glycolipid-anchored membrane

protein with an extracellular binding domain for LPL; LPL forms a 1:1 complex with GPIHBP1[8,9]. GPIHBP1 transports LPL into the capillary, where it can interact with triglyceride rich lipoproteins (TRLs) to release FFAs[10]. LPL can also be found free from GPIHBP1 and associated with mobile TRLs in the blood[11]. Independent from its role hydrolyzing triglycerides, LPL on TRLs mediates lipoprotein remnant uptake by the liver[12,13].

Historically, LPL was thought to have two oligomeric states: an active dimer and inactive monomer[14,15]. Recent structural work has challenged this paradigm. First, two crystal structures of LPL in complex with GPIHBP1 were solved[8,9]. The LPL/GPIHBP1 heterodimer formed a heterotetramer in the asymmetric unit of the crystal, but this conformation occluded access to the active site and further analysis suggested it was likely an effect of crystallization[8]. Size exclusion multiangle light scattering (SEC-MALS) was used to show that in solution LPL/GPIHBP1 exists as a 1:1 heterodimer[8]. This work confirmed that the LPL/GPIHBP1 heterodimer is an active form of LPL. It was also observed that monomers of LPL may be active in the absence of GPIHBP1[16]. Next, an inactive helical oligomer of LPL was identified by cryoEM[5]. These studies left the structure of the LPL dimer unresolved. Studies using radiation inactivation, ultracentrifugation, enzyme-linked immunosorbent assay (ELISA) sandwich assays, and single-molecule fluorescence resonance energy transfer (smFRET) indicate the existence of an

[1]Department of Biochemistry and Biophysics, University of North Carolina, Chapel Hill, NC 27599, USA. ✉e-mail: neher@email.unc.edu

LPL dimer[14,15,17–19]. A shared feature of these experiments that showed the existence of dimeric LPL was that they were performed without GPIHBP1.

Structures of many mammalian lipases have been solved, including pancreatic triacylglycerol lipase[20–22], pancreatic lipase-related protein 1[23] and 2[24], monoacylglycerol lipase[25], lipoprotein lipase[5,8,9], gastric lipase[26], bile-salt activated lipase[27,28], and lysosomal acid lipase[29]. A few of these structures were solved in open conformations where the lid peptide residues were shifted to expose a hydrophobic pocket adjacent to the active site[21,22,24]. Lipases achieve an open state by undergoing interfacial activation, which occurs when the lipase associates with a nonpolar-aqueous interface[30]. The combination of the lid peptide and hydrophobic pocket is thought to provide mammalian lipases substrate selectivity. An outlier in the eukaryotic lipase family is found in fungi, specifically *Candida rugosa* lipase[31], which uses a tunnel for substrate binding rather than a hydrophobic pocket. Structures have shown acyl chain analogs are able to enter and occupy this hydrophobic tunnel[31]. Subsequent mutational studies have shown that mutations made in this tunnel can alter substrate specificity[32].

We set out to solve the structure of LPL in the absence of GPIHBP1, hypothesizing that this would reveal an active LPL dimer. We succeeded in using cryoEM to solve the structure of a dimeric LPL oligomer. The dimeric LPL structure further highlights the oligomeric diversity of LPL, as now LPL homodimer, heterodimer[8,9], and helical filament[5] structures have been elucidated. The interaction interfaces for these three oligomers share overlapping residues, making it clear that they are mutually exclusive states. The structure also reveals the presence of a hydrophobic pore spanning the N-terminal domain of LPL, adjacent to the active site, suggesting it plays a role in acyl chain hydrolysis. The prevailing theory prior to this structure was that an open lipase conformation was defined by a displaced lid peptide, exposing the hydrophobic pocket surrounding the active site. After the lid opens, the substrate enters the active site, is hydrolyzed and then released in a bidirectional manner. Based on our structure, we propose a model for lipid hydrolysis, in which the free fatty acid product travels unidirectionally through the active site pore entering and exiting opposite sides of the protein. The pore may also provide additional substrate discrimination. Structural similarity of LPL to other human lipases suggests that the existence of a hydrophobic pore could be conserved but has not been observed due to the difficulty of studying lipase structures in the presence of an interfacial substrate.

## Results

### LPL treated with deoxycholate prevents formation of inactive helices

To generate a homogenous sample of active LPL for cryoEM, we needed to disrupt the inactive LPL helices that form at high protein concentrations. We previously found that treatment with the bile salt deoxycholate causes dissolution of LPL helices[5], and LPL treated with deoxycholate is stable and active[33]. Therefore, we treated LPL with deoxycholate and dialyzed it into buffer both with and without additional deoxycholate. Deoxycholate has a critical micelle concentration (CMC) that can vary based on temperature, but the concentration we used (1 mM) is below the CMC and can be removed by dialysis[34]. We analyzed both conditions by negative stain transmission electron microscopy (nsTEM) and cryoEM and found LPL helices were not present in either. These data show that binding to deoxycholate and helix dissolution is not affected by subsequent dialysis to a buffer without deoxycholate. We also tested the activity of LPL by monitoring the release of FFA from VLDL using an enzyme-coupled fluorescent assay[35]. Interestingly, LPL activity was slightly diminished when deoxycholate was present compared to LPL alone. We found that the activity of deoxycholate-treated LPL subsequently dialyzed with or without deoxycholate was comparable (Supplementary Figs. 1A, B). Bovine LPL, which shares ~94% sequence identity with human LPL[36],

was used for the experiments described in this work unless otherwise specified.

### LPL forms a C-terminal to C-terminal homodimer

After confirming we had a homogenous, active preparation of LPL (Supplementary Fig. 1C), we analyzed the sample using single particle cryoEM (Supplementary Fig. 2A). Initial 2D classification revealed one class of LPL with clear secondary structure visible (Fig. 1a). However, few other classes were present in the data. This result suggested that LPL was adopting a preferred orientation by interacting with the air/water interface (AWI) during grid freezing[37,38]. Multiple methods were attempted to alleviate the preferred orientation via detergents and additives. None of these were able to disrupt the association of LPL with the AWI. We ultimately were able to overcome the preferred orientation despite the relatively small size of our protein (~100 kDa dimer) by using a tilting strategy (Fig. 1a) combined with specific enrichment of rare particle views using the Topaz neural network[39] (Supplementary Fig. 2b). We collected data sets at 3 tilts: 0°, 30°, and 45°. We processed each tilted data set individually using cryoSPARC[40] and focused on enriching for rare particle orientations by iteratively training Topaz[39] to pick non-preferred orientation particles for each tilt set. We then combined the particles from all tilts.

We found that the LPL dimer had C2 symmetry, which was applied after initial rounds of C1 symmetry refinement (Fig. 1b). The final resolution was 3.9 Å using the gold-standard Fourier shell correlation (GSFSC) with a cut-off of 0.143 on the directional FSC (Fig. 1d). Analysis with the 3DFSC server[37] indicated that although anisotropy was still present in the structure, the sphericity was 0.85 out of 1, highlighting the success of a combined tilting and Topaz picking strategy. The distribution of particles from each view of the final structure is illustrated in Supplementary Fig. 3a. We were able to unambiguously dock LPL into the density, identify secondary structure, observe separation of β-sheets (Supplementary Fig. 2C), and see density for large amino acid side chains. A map of LPL was built into the density using Coot and Phenix (Fig. 1e, Table 1). LPL has a flexible lid peptide that covers its active site, and we were unable to build the map to fit the LPL lid density at 3.9 Å, although it can be clearly seen when the thresholding is reduced (Fig. 1c). The correlation between the map and model is 0.77 (Table 1). Due to the residual anisotropy of the model, there is lower confidence for the fit of some amino acid side chains, leading to the lower correlation value. The interior residues of the protein were at higher resolution than the outer residues (Supplementary Fig. 3B).

The structure shows that LPL forms a homodimer associated at the C-terminal domains ("tails") of both subunits (Fig. 1). Intriguingly, in the tail-to-tail dimer, two features of LPL which are key for substrate interaction are situated along the same surface. Specifically, the C-terminal tryptophan-loop, which helps recognize TRL substrates[41], and the N-terminal lid peptide, which opens in response to a lipid-water interface, are arrayed on a single plane (Fig. 1c).

### Active site pore in the LPL dimer structure

The active LPL dimer is oriented with both its substrate recognizing tryptophan loops and active sites facing the same direction, suggesting an active conformation that could be engaged with a TRL substrate (Fig. 2a). We thus analyzed the LPL active site in the structure and found a clearly defined pore traversing LPL which is directly adjacent to the active site residues. The fit of the pore lining residues to the cryoEM map can be seen in Supplementary Fig. 4. This open pore in LPL is facilitated by shifts of many amino acids, these differences are illustrated by morphing other LPL structures to the LPL from our homodimer (Supplementary Movie 1–3).

We used MOLEonline[42], an automated toolkit for analysis of tunnels and pores, to identify the active site pore using our protein model. A pore with a length of ~45 Å was identified that passes through the N-terminal domain of LPL (Fig. 2b–d). We also performed this analysis

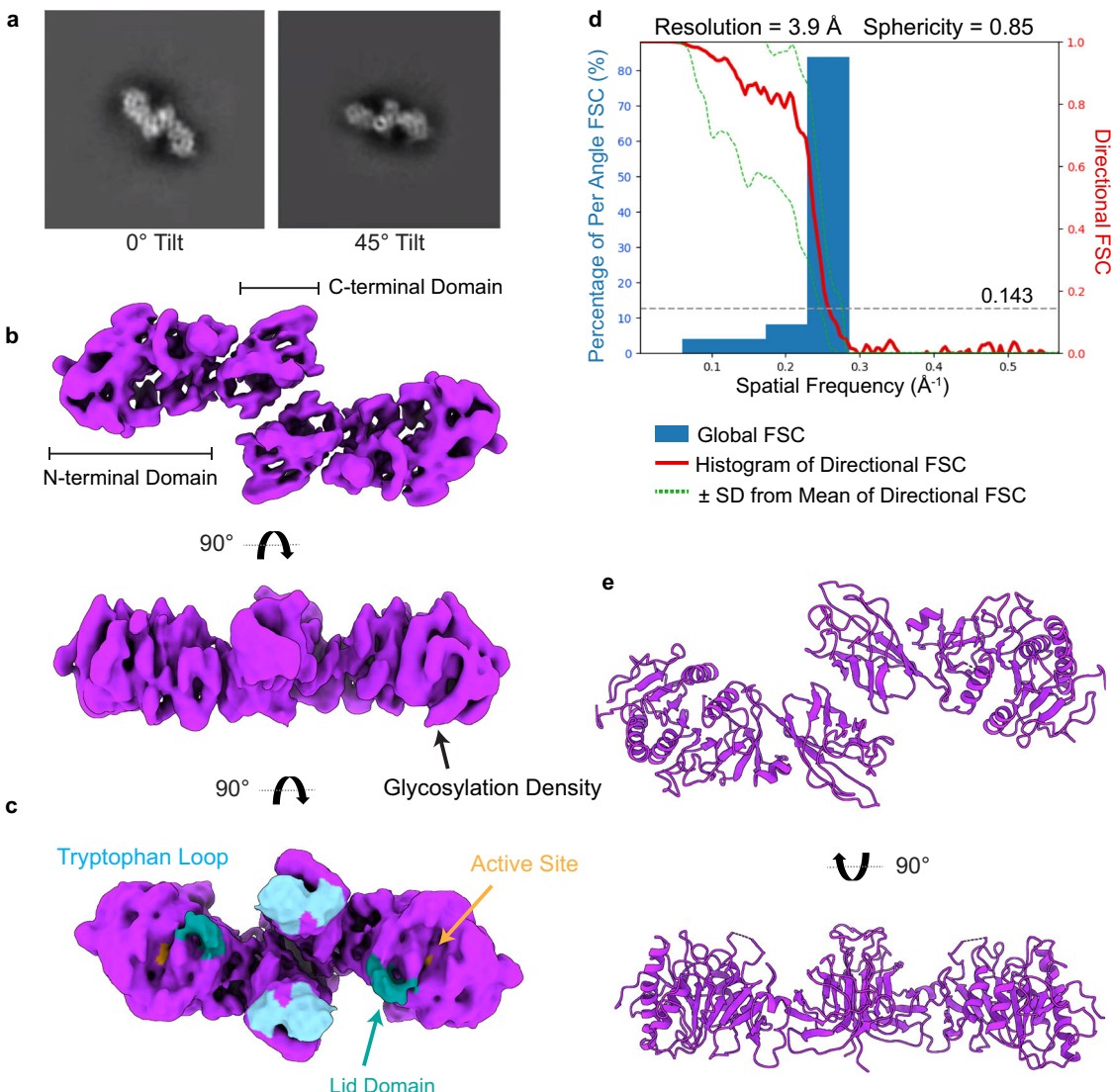

**Fig. 1 | LPL forms C-terminal to C-terminal homodimers. a** The 2D class of LPL's preferred orientation at 0° (left). Tilting the grid to 45° allowed us to capture an alternate view of this orientation (right). **b** The density map for the LPL dimer (purple). The dimerization interface is between the C-terminal domains of both LPL subunits. Density corresponding to the glycosylation of N73 was observed. **c** When thresholding of the density map is increased, density for the lid domain of LPL appears (aqua). The tryptophan loop in the C-terminal domain, which is involved in substrate recognition, can be seen along the same plane as the density corresponding to the lid peptide (cyan). The lid peptide is likely too mobile to be resolved at high resolution. The lid domain density has shifted downward and to the side of the active site (orange) adopting an open conformation. **d** The resolution of the LPL dimer is 3.9 Å with a sphericity of 0.85 out of 1, which indicates that we were able to create a nearly complete reconstruction, despite anisotropy resulting from the preferred orientation of the particles (Supplementary Fig. 3B). Source data are provided as a Source Data file. **e** The molecular map of the LPL dimer was built into the density map (Supplementary Fig. 4E). The residues of the lid domain were not modeled because their location was not determined with high resolution.

on LPL from the helical LPL oligomer, LPL/GPIHBP1, and other mammalian lipase structures. No comparable pores were identified. The pore in the dimer structure has a bottleneck with a radius of 1.4 Å, which is sufficient for the carbon hydrogen backbone of a fatty acid. The pore has a highly hydrophobic section that could accommodate a fatty acid tail, potentially aligning the hydrolysable ester bond with the active site residues in a relatively more hydrophilic section of the pore entrance (Fig. 2c, d). We looked at hydropathy of the pore, which assigns hydrophilic amino acids a negative value and hydrophobic amino acids a positive value to examine the overall environment of the pore[43]. We found that the inversion from negative to positive hydropathy was located at the LPL active site residues (Fig. 2c). At the top of the pore, where we would expect the glycerol backbone to be located during glyceride hydrolysis, there is a negative hydropathy score, indicating hydrophilicity. This transitions to a positive score following the active site, suggesting that the hydrophobic fatty acid chain could

rest in the remainder of the pore. A 16-carbon fatty acid, such as palmitate, would have a length of ~24 Å and the ~30 Å of pore following the LPL active site is more than sufficient to hold that length of hydrophobic molecule. To better understand the ability of acyl chains to fit into the hydrophobic pore we modeled 3 separate ligands into the pore using PyRosetta[44], palmitate, oleate, and lineolate (Fig. 2e). The pdb files for these models are available as Supplementary Data 1–6. We found that both saturated and unsaturated ligands could be accommodated in the pore. We also modeled a triglyceride into the structure (Supplementary Fig. 5, Supplementary Data 7–8).

### Comparison of active LPL to other open lipase structures

We next compared our structure to other known lipase structures, including structures of pancreatic triacylglycerol lipase (PTL) with the lid peptide in an open position[22,30,45] (Fig. 3). PTL shares significant structural and mechanistic homology with LPL, as both have lid

**Table 1 | CryoEM Validation Statistics**

| Data Collection and Processing | |
|---|---|
| Magnification | 45,000 |
| Voltage (kV) | 200 |
| Pixel size (Å) | 0.88 |
| Total dose (e⁻/Å²) | 55.1 |
| Number of particles | 527,205 |
| B-factor | –100 |
| **Map resolution (Å)** | |
| Model:map FCR (0.5) | 4.5 |
| Map:map FSC (0.143) | 3.9 |
| $d_{99}$ | 4.6 |
| Symmetry | C2 |
| **Model refinement** | |
| Model to map CC | 0.77 |
| Clash score, all atoms | 4.84 |
| Ramachandran favored (%) | 90.51 |
| Ramachandran outliers (%) | 0 |
| Rotamer outliers (%) | 0.27 |
| $C_\beta$ deviations > 0.25 Å | 0 |
| **RMS deviations** | |
| Bond (Å) | 0.005 |
| Angles (°) | 1.277 |
| **Deposition IDs** | |
| PDB ID | 8ERL |
| EMDB ID | 28,554 |

peptides that regulate access to the active site and swing open during interfacial activation. Two of the open PTL structures feature hydrophobic ligands bound in the hydrophobic pocket, phosphatidyl choline and tetraethylene glycol monooctyl ether (TGME), respectively (Fig. 3f–i). The lid peptides in these two PTL crystal structures and our LPL dimer have shifted into an open state, exposing the active site (Fig. 3c, g, i). However, a cross-sectional comparison of both PTL structures with the LPL dimer shows a substantial difference between the size of the previously characterized hydrophobic pocket and the hydrophobic pore seen in our LPL structure (Fig. 3b, f, h). The hydrophobic pore is unique to our structure among known mammalian lipases.

## The LPL dimerization interface resembles the GPIHBP1 binding interface and helical LPL interface

We next examined the C-terminal to C-terminal LPL dimerization interface (Fig. 4a). Prior structural studies show that LPL has multiple possible interaction interfaces as observed with GPIHBP1 and the LPL helix (Fig. 4a–c). Comparison of these interaction interfaces revealed that the LPL dimerization interface shared almost complete overlap with the GPIHBP1 binding interface (Fig. 4d, e). To further explore this shared binding interface, we analyzed it using PDBePISA[46]. The interface involves 14 amino acids from each LPL subunit, and 12 of these residues are shared with the LPL/GPIHBP1 interface (Supplementary Data 9). The homodimerization interface also overlaps with interfaces involved in formation of the inactive LPL helix. The base subunit of the LPL helix is an inactive dimer, which interacts with other inactive LPL dimers through helical interfaces to form a helical filament[5]. There was no overlap between the inactive helical dimerization interface and the active homodimer interface (Fig. 4g). However, there was overlap with the other helical interfaces, they share 13 interacting residues (Fig. 4f). Of these 13 residues, 3 were involved in the C-terminal to C-terminal helical interface, and 11 in the C-terminal to N-terminal helical interface

(one residue interacts with both). This suggests that active dimers are incompatible with formation of the repeating helical structure because the helical interaction interfaces are occluded by the presence of the other LPL subunit in the helical interface. All three interfaces—LPL homodimer, LPL/GPIHBP1, LPL helix—share residues Y-G-T-V-A-E (370–375 in bovine LPL and 367–372 in human LPL) and L406, M407, and K409 in bovine LPL (403, 404, and 406 in human LPL). This patch is hydrophobic due to the presence of tyrosine, valine, leucine, alanine, and methionine.

## LPL oligomeric state is influenced by concentration and additives

We next set out to use an independent method to assess the variability in LPL's oligomeric state. We did so in part because deoxycholate was needed to prevent LPL helix formation and bile acids are present only at low levels in the blood (~350 nM of deoxycholate[47]) and display high interindividual variability[48]. We wanted to ensure that the higher levels of deoxycholate used to break up helices did not alter the formation of LPL oligomers in solution. First, we used mass photometry to characterize the molecular weight of LPL at low concentrations. Mass photometry is an emerging technique that detects the impacts of protein particles hitting a glass coverslip to determine the molecular weight of the proteins in solution[49]. This technique allows for analysis of relatively low concentration protein samples in solution.

At 16 nM LPL, the molecular weight was best fit by a single Gaussian distribution (Fig. 5a) with a peak at 56 kDa, correlating to a monomer of LPL (50.5 kDa). When the concentration of LPL was increased, a shift toward higher molecular weight was observed (Fig. 5a). At higher concentrations (50 and 78 nM) we observe a higher molecular weight shoulder appearing on the monomer peak, suggesting that a second oligomeric species is beginning to form. We fit these concentrations with multiple Gaussian distributions, although we do not believe that the molecular weight suggested by the secondary peak (77 and 73 kDa respectively) is an accurate reflection of the exact molecular weight, since it is a subspecies with significantly fewer data points compared to the monomeric peak. However, when we increase the concentration to 100 nM, we see a peak at 99 kDa formed by the majority of the particle impacts, suggesting that at this concentration LPL has transitioned from being primarily monomeric to majority dimeric. At 100 nM we again see a shoulder forming (131 kDa), perhaps indicating that another higher weight oligomer is forming.

When LPL at 16 nM LPL was measured in solution containing 0.25 mM deoxycholate, we again saw a single peak of monomeric LPL at 63 kDa (Fig. 5b). As we increased LPL concentration in the presence of deoxycholate we saw a similar pattern to LPL alone, with a higher molecular weight shoulder being observed at 50 nM (81 kDa). At 100 nM of LPL with deoxycholate, we still see two peaks, but the primary peak has shifted to a mass indicative of LPL dimerization: 97 kDa. This matches the data for 78 nM LPL with deoxycholate, which also shows two peaks, although with lower occupancy, at 99 kDa. When fit with multiple Gaussians, we see there are still contributions from lower molecular weight species at 78 nM and 100 nM (81 kDa and 72 kDa respectively), but as the concentration of LPL increases the ratio of the dimeric protein to the lower molecular weight protein increases, suggesting a correlation between the increasing concentration and the population of LPL dimers.

We also tested an additional LPL additive, heparin, which binds to LPL and has been shown to stabilize higher order oligomers[5]. We mixed 27 U/mL mixed molecular weight heparin with LPL and at 16 nM LPL observed the appearance of a single molecular weight species at 66 kDa (Fig. 5c). The LPL monomers observed with deoxycholate and heparin have a slightly higher molecular weight than LPL alone, this may be due to association with heparin or deoxycholate increasing the apparent molecular weight. At higher LPL concentrations of 50 nM and

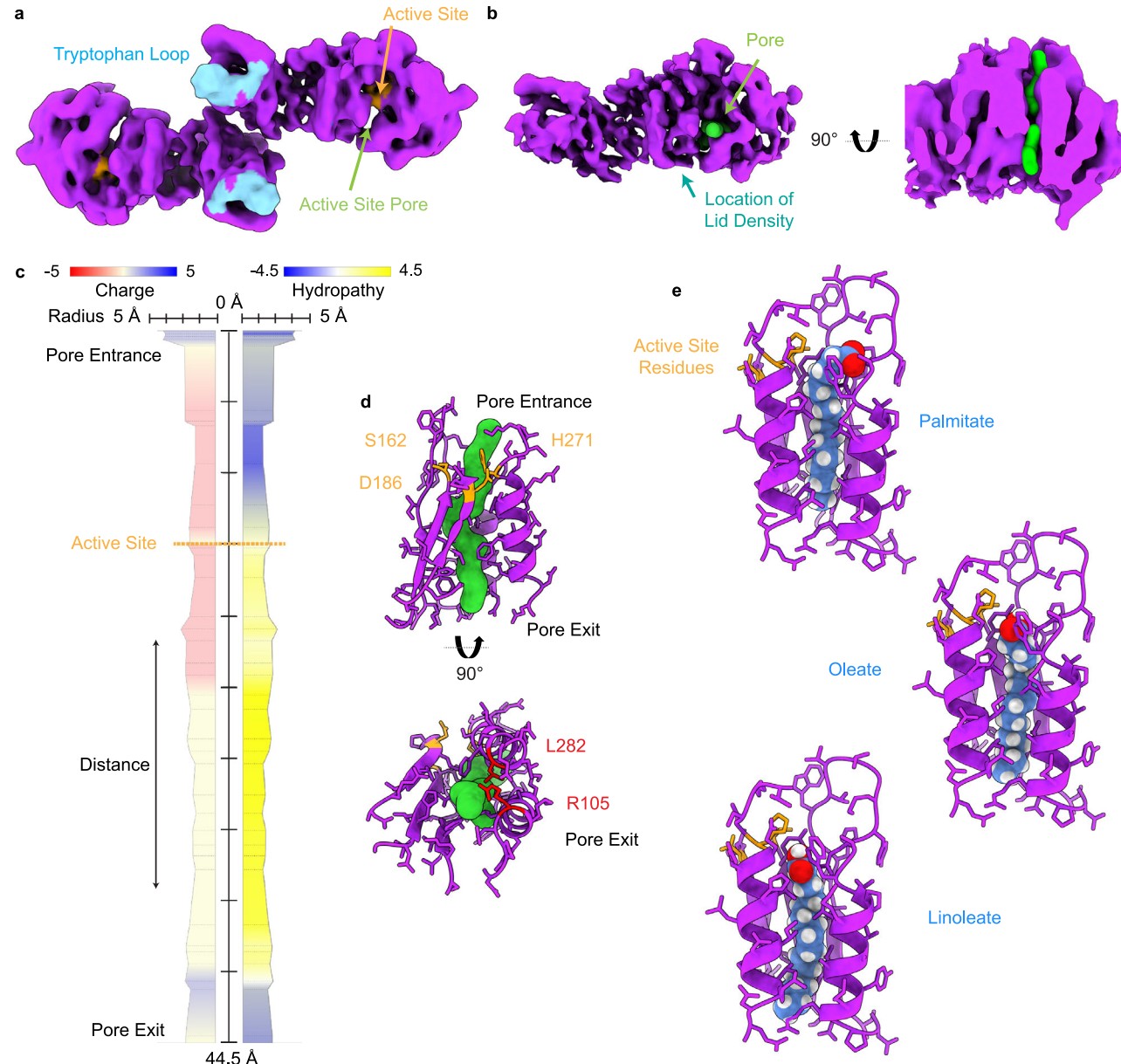

**Fig. 2 | The LPL homodimer reveals an open active site pore. a** The LPL dimer map colored to show the LPL active site residues (orange) and the tryptophan loop (cyan), which is involved in substrate recognition. An area without protein density is seen in the N-terminal of LPL suggested the existence of a pore spanning the protein. **b** We analyzed the LPL model and were able to identify a pore (green) that spanned the N-terminal domain, shown here in the density map. The pore is located adjacent to the active site and an aqua arrow indicates the position of the lid density if thresholding were lower to the side of the pore. A cutaway of the density shows that the pore spans the N-terminal domain of LPL. **c** MOLEonline was used to create a trace of the characteristics of the amino acids that line the identified pore. The residues corresponding to the entrance to the LPL pore are at the top of the diagram and the path of the pore travels to the terminus at the bottom. The pore is 44.5 Å in length with a bottleneck radius of 1.4 Å. The location of the active site is marked by a dotted orange line. Charge and hydropathy colors correspond to the legends above the diagram. The source data for this graph are provided as a Source Data file. **d** The residues surrounding the pore are displayed to show how the pore interacts with surrounding amino acids. The active site residues (S162, D186, and H271) are indicated in orange. H271 directly abuts the pore density. Looking at the pore terminus (the end opposite of the LPL lid domain), we can see that there is an unobstructed exit from the pore. Two residues, L282 and R105 (red), known to cause LPL deficiency in humans are located near the pore terminus. Models of these mutations can be seen in Supplementary Fig. 9. **e** Three common acyl chains were fit into the hydrophobic pore using PyRosetta. Ligands shown as sphere representations are saturated fatty acid palmitate (C16:0), monounsaturated oleate (C18:1), and doubly unsaturated lineolate (C18:2) (carbon atoms are blue, hydrogen white, and oxygen red). The active site residues are again shown in orange. The starting and final models for these fits are available as Supplementary Data 1–6.

78 nM, we again observed a shift to multiple Gaussian distributions, with minor peaks at 109 and 100 kDa respectively, which correlate to LPL dimers. Interestingly, in the heparin condition the molecular weight matching a dimer appears at a lower concentration than with LPL alone or deoxycholate. We also see peaks for a lower molecular weight species (83 and 70 kDa respectively). The general distributions at both 50 and 78 nM appear similar, however, at 100 nM we see a

complete transition to dimeric oligomers (109 kDa), with a shoulder indicating a higher order molecular weight species (131 kDa). This data indicates that the oligomerization of LPL is dynamic with respect to concentration, but also that deoxycholate is not significantly affecting the ability of LPL to move through these oligomeric states.

We also confirmed the concentration dependence of oligomer formation seen with bovine LPL is shared by human LPL, by analyzing

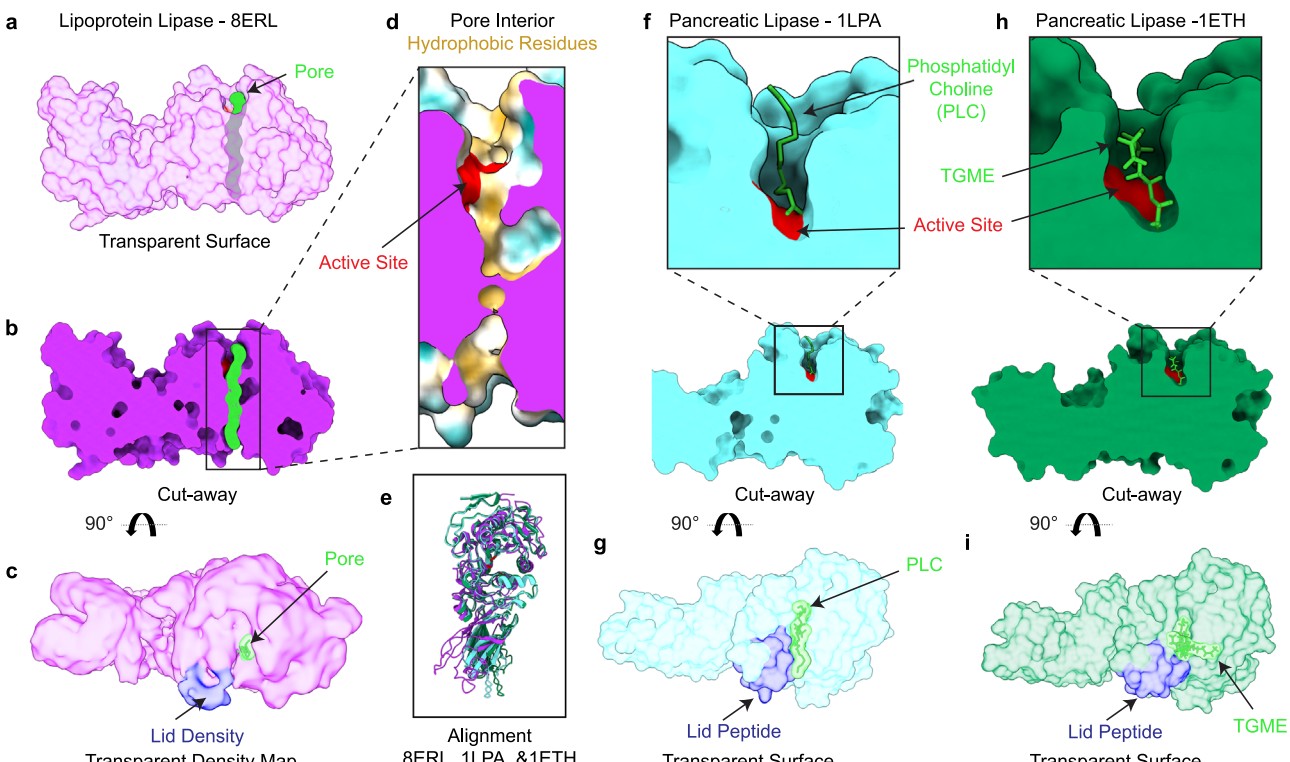

**Fig. 3 | A pore is not present in other open lipase structures. a** A transparent surface representation of a single subunit of LPL (purple) from the LPL dimer, with the path of the pore (lime green). **b** A cutaway shows the path of the pore through the protein. **c** The density corresponding to the LPL lid domain (blue) has swung to the side giving access to the active site and pore. **d** A closeup look at the LPL pore cutaway, with the active site residues (red) and other residues colored on the surface to show hydrophobic (yellow) and hydrophilic (aqua) character. The cut-away surface is in purple. **e** Alignment of LPL and two pancreatic triacylglyceride lipase (PTL) structures (cyan (1LPA) and green (1ETH)) shows similar folds and well aligned location of the active site residues (red). **f** A cutaway of the structure of human PTL (cyan) aligned to LPL – no pore is observed, only a small pocket adjacent to the active site (red). The PTL/procolipase complex was crystallized in the presence of phophatidyl choline micelles and bile salt (PDB 1LPA). A zoom-in of the active site in the cross-section shows a diundecyl phosphatidyl choline (PLC) bound (lime green). **g** PTL in the presence of procolipase was observed with an open lid peptide (blue). **h** A cutaway structure of porcine PTL (green) from a PTL/colipase complex, solved in the presence of bile salts and detergents, shows the active site (red) and a tetra ethylene glycol monooctyl ether (TGME) ligand (lime green) is observed bound in the hydrophobic pocket (PDB 1ETH). **i** A transparent surface shows that the lid peptide (blue) is in an open state, providing access to the hydrophobic pocket for TGME (lime green) binding.

both 16 nM and 50 nM human LPL (Supplementary Fig. 6A). At 16 nM human LPL has a primary peak at 58 kDa that primary peak shifts to 99 kDa at 50 nM.

## LPL forms multiple higher order oligomers

Mass photometry is an effective technique for dilute solutions, but we also wanted to capture LPL oligomers from higher concentration solutions and observe any differences in LPL oligomer formation. We used gradient fixation (GraFix) to stabilize oligomers for later analysis[50,51]. We separated both 1 μM LPL and 1 μM LPL with 1 mM deoxycholate using a linear glycerol gradient, which also contained a gradient of glutaraldehyde crosslinker, resulting in individual fractions that contain <100 nM of protein. GraFix allows gentle crosslinking, while simultaneously separating the resulting oligomers by size. GraFix is a common method for cryoEM that is not known to induce formation of non-specific oligomers[50]. We assayed the resulting crosslinked proteins by both western blot and TAMRA-FP serine-hydrolase active site probe (Supplementary Fig. 7). We found that LPL with and without deoxycholate form the same array of oligomers in solution during GraFix with 4 clear bands (Fig. 6a, b). Because crosslinking of LPL oligomers alters their electrophoretic mobility (i.e. a crosslinked monomer of LPL runs at a lower molecular weight than an uncrosslinked LPL monomer), we used mass photometry to confirm the molecular weight of the LPL oligomers observed in the different fractions (Fig. 6c, d). Further, by averaging data from multiple fractions, we could confirm the identities of monomers, dimers, trimers, and tetramers of LPL by molecular weight (Fig. 6e). We also confirmed that human LPL when crosslinked in solution results in the appearance of the same four oligomeric species as bovine LPL (Supplementary Fig. 6b).

In contrast to the solution crosslinking data we did not observe monomers or trimers in our cryoEM data. However, in addition to LPL dimers we did observe some 2D classes that were apparent tetramers of LPL (Supplementary Fig. 8A). The tetramer formed when two LPL dimers came together. One LPL subunit of each oligomer appeared to rotate, forming a head-to-tail interaction that may mimic the inactive dimer, which is the repeating subunit of the LPL helix[5] (Supplementary Fig. 8B, C).

## Discussion

LPL has long been known to form an active dimer[14,15,17–19] and our cryoEM structure reveals the molecular architecture of this complex, including an unexpected C-terminal to C-terminal interaction interface. LPL was previously predicted to form a head-to-tail (N-terminal to C-terminal) dimer[17,52,53]. This head-to-tail arrangement was observed in the inactive LPL helix and the heterotetramer of the LPL/GPIHBP1 crystal structures. However, in our structure LPL forms a tail-to-tail dimer. Analysis of the homodimerization interface shows significant overlap with the LPL/GPIHBP1 interface and the LPL helical interfaces. This suggests that LPL homodimerization, binding to GPIHBP1, and LPL helix formation are all mutually exclusive states. Therefore, when LPL is bound to GPIHBP1, it is not able to form a dimer, but upon

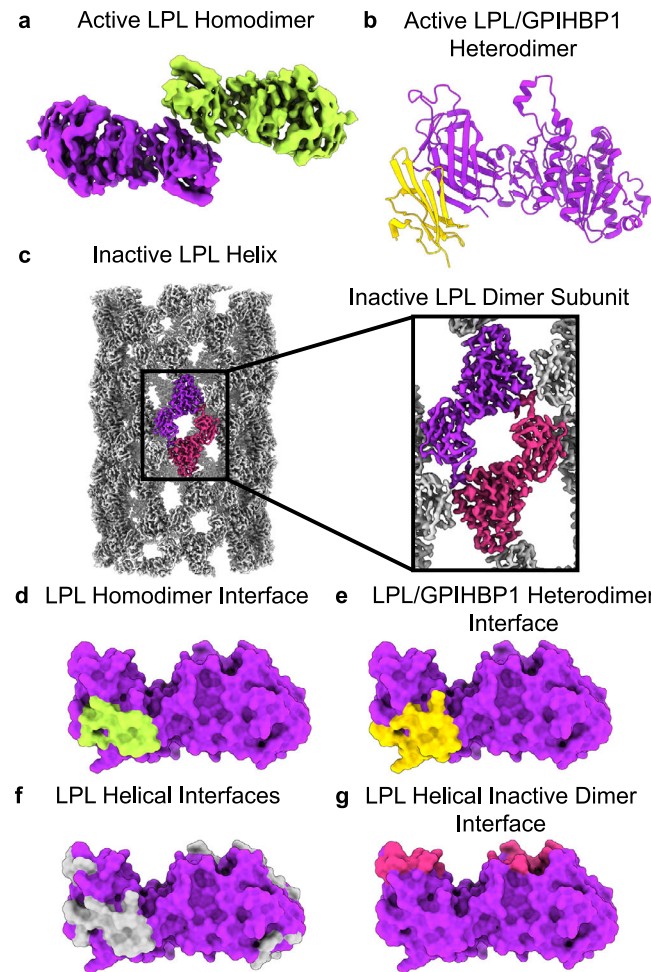

**a** Active LPL Homodimer
**b** Active LPL/GPIHBP1 Heterodimer
**c** Inactive LPL Helix

Inactive LPL Dimer Subunit

**d** LPL Homodimer Interface
**e** LPL/GPIHBP1 Heterodimer Interface
**f** LPL Helical Interfaces
**g** LPL Helical Inactive Dimer Interface

**Fig. 4 | The LPL dimerization interface overlaps with other known interfaces, including LPL/GPIHBP1 and the LPL helix.** The structure of LPL (purple) has been solved in 3 distinct states. **a** An active homodimer described in this work (one LPL subunit is colored lime green), **b** a heterodimer with GPIHBP1 (gold) (shown as PDB 6OB0), and **c** an inactive helix (EMDB 20673), for which the repeating subunits are inactive LPL dimers (purple and magenta, the rest of the LPL in the helix is colored gray). We used PDBePISA to identify the interacting residues for each of these complexes. This analysis is available as Supplementary Data 9. **d** In the LPL homodimer the interaction interface was located at the C-terminal, and we mapped the residues identified as being buried by the interface onto a space filling representation of one monomer of the LPL homodimer (interface residues, colored lime green). **e** The LPL residues interfacing with GPIHBP1 (**e**, gold) are mapped onto the same LPL, as are the helical interfaces (**f**, gray) and the inactive dimer interface (**g**, magenta). The helical interfaces represent the interactions that occur in the LPL helix that are not with the companion LPL subunit found in the inactive dimer. There is an overlap between the active LPL homodimer, LPL/GPIHBP1 heterodimer interface, and LPL helical interfaces.

dissociation from GPIHBP1, LPL would be able to form a homodimer. This C-terminal interaction site shared in all three known LPL structures[5,8,9] may play a role in stabilizing LPL, whether in an active or inactive form, enabling structural characterization. The C-terminal to C-terminal arrangement also allows the tryptophan loop, lid peptide, and active site residues of both LPL subunits to interact with substrate at the same time, which was not seen in previous models of the LPL dimer[52].

We also observed a hydrophobic pore adjacent to the LPL active site, formed by rearrangement of the residues lining the pore. The closest analog we found in the literature to this open pore is the tunnel found in *C. rugosa* lipase, in which the tunnel gates entrance to the active site[31]. However, the *C. rugosa* lipase has not been shown to have

a separate exit as we see in the LPL pore. Our analysis of the LPL pore revealed a pore ideally suited to harbor the hydrophobic fatty acid tail of a triglyceride. The hydrophobic pore could provide an additional layer of substrate discrimination for LPL beyond the previously identified lid peptide and hydrophobic binding pocket. In that regard, the pore might utilize a similar specificity mechanism to the tunnel seen in *C. rugosa* lipase and other homologs[54,55]. This could provide an explanation for why LPL preferentially hydrolyzes the sn-1 and sn-3 position on triglycerides, as in our modeling the sn-2 acyl chain interacts in a separate hydrophobic pocket, which could facilitate enhanced substrate recognition of the sn-1 or sn-3 position by the hydrophobic pore (Supplementary Fig. 5). Overall, ligand specificity for LPL would then be conferred by a combination of the lid peptide, hydrophobic pocket, and hydrophobic pore.

The pore layout also suggests the existence of a unidirectional mechanism where the FFA product exits from the bottom of the pore, which would allow the FFA to enter the capillary, while the remaining monoacyl- or diacyl-glycerol returned to the chylomicron (Fig. 7). This mechanism brings previously known substrate binding regions together and integrates them with the existence of an active site pore. Further, once a triglyceride is hydrolyzed, the diacyl-, monoacyl-glycerol, or glycerol is then able to exit from the entrance of the pore and diffuse back to the surface layer of the TRL[56], while the liberated FFA can pass unidirectionally through the pore and directly into the capillary where it can bind to albumin for transport to the capillary endothelium[57,58]. This mechanism has the appeal of being very efficient, as LPL would not be required to repeatedly undergo interfacial activation in order to bind to a new ligand or to dissociate from the TRL to release FFA. The processive hydrolysis of ligands by unidirectional travel of the FFA products could also be a mechanism used by other mammalian lipases.

Previous work on the *C. rugosa* lipase tunnel has shown the mutations in the tunnel can tune the substrate specificity of the lipase[55]. Mutations leading to LPL deficiency have been extensively characterized to better understand the causes of familial hyperchylomicronemia. We therefore decided to compare the amino acids that line the pore in LPL to known mutations resulting in LPL deficiency and look at their influence on substrate specificity. We identified R102S in humans (R105 in bovine structure), which leads to LPL deficiency[59,60], and L279R in humans (L282 in bovine structure), which leads to lower LPL specific activity[61]. These two mutations which are both known to have deleterious effects on human LPL activity localize to the pore exit (Fig. 2d). (In literature predating structures of LPL, these residues were referred to as R75 and L225 in humans, accounting for removal of the signal sequence. We will refer to these residues by their numbering in recent LPL structures for consistency.) The LPL R102S variant was identified as one of two mutations in a patient with compound heterozygous loss of LPL function resulting in severe chylomicronemia[60]. The R102S mutation results in loss of positive charge at the pore exit (Fig. 2d and Supplementary Fig. 9A, C). In previous work LPL with the R102S mutation was expressed in and purified from cultured cells, and a reduced amount of active enzyme was produced relative to the WT LPL. Interestingly, when corrected for protein concentration, the WT LPL showed higher specific activity on a long chain, triolein substrate relative to LPL R102S, but LPL R102S showed higher specific activity on the small soluble substrate, para-nitrophenyl butyrate[60]. The LPL L279R mutation substitutes a hydrophobic leucine residue with a bulky, positively charged arginine residue. Residue 279 is positioned directly inside of the pore close to the exit (Fig. 2d). Homology modeling in our structure suggests that this bulkier arginine residue may block the pore (Supplementary Fig. 9B, D). A patient heterozygous for the L279R mutation developed severe chylomicronemia during pregnancy[61]. In vitro studies revealed that arginine or proline substitutions at LPL residue 279 resulted in production of 30% or 70% of WT protein, respectively. However, both LPL variants had no specific

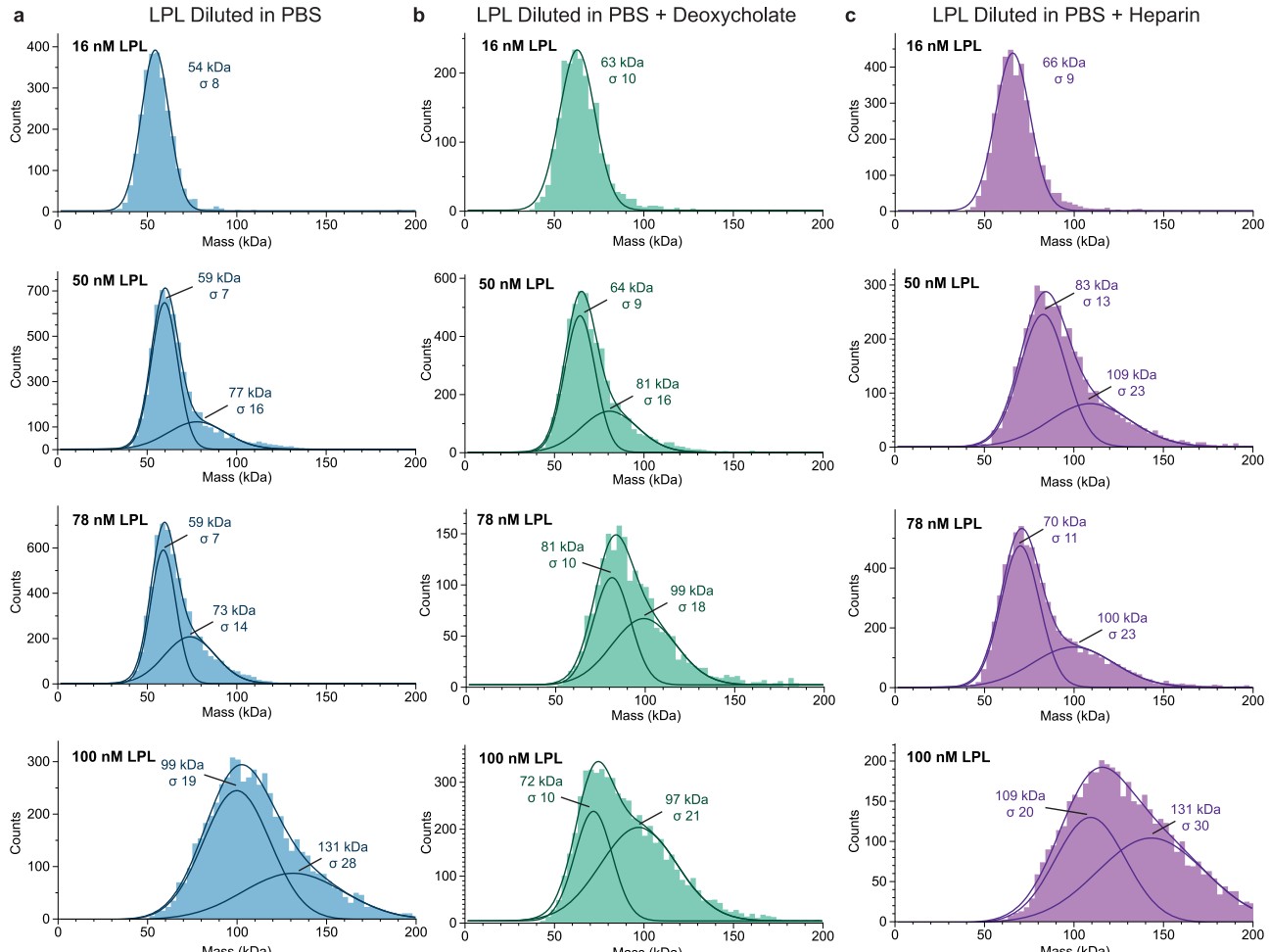

**Fig. 5 | Concentration Drives LPL oligomerization. a** Mass photometry of LPL (blue) at increasing concentrations shows that when LPL is analyzed at low concentrations it forms a monomer, and as LPL concentration increases a higher molecular weight shoulder appears. At 100 nM LPL the primary peak corresponds to the molecular weight of an LPL dimer. **b** LPL in PBS with 0.25 mM deoxycholate (green) displays a similar trend to LPL without deoxycholate. **c** LPL measured in PBS with 27 U/mL heparin (purple) also shows the shift to dimerization with increasing concentration. However, the appearance of a dimeric molecular weight peak is observed at lower concentration (50 nM) compared to LPL or LPL with deoxycholate. The data was fitted with single or multiple Gaussian distributions to determine an average molecular weight (kDa) and the width of the standard deviation of the fitted Gaussian is given by sigma (σ). The combined trace of the multiple Gaussians is also shown to illustrate the fit of the Gaussians to the histograms. LPL monomer theoretical molecular weight = 50.5 kDa. Source data are provided as a Source Data file.

activity with a long chain triglyceride substrate, specifically a triolein emulsion[61]. Thus, both LPL pore mutants identified from chylomicronemic patients could fold and be secreted. However, both variants showed a specific defect on long chain triglyceride substrates, suggesting the pore observed in our structure has physiological relevance for hydrolysis of specifically long chain ligands. Whether these mutations affect release of LPL product or alter substrate specificity requires further investigation.

The LPL dimer structure was solved in the presence of deoxycholate to create a homogenous sample. However, the LPL dimer does not form solely because of deoxycholate treatment. The appearance of the LPL dimer is linked to increasing LPL concentration. We used crosslinking to capture LPL dimers in samples both with and without deoxycholate. We observe that the distribution of oligomers captured in both solutions are similar, with monomers, dimers, trimers, and tetramers observed by SDS-PAGE and mass photometry. It has been previously reported that other lipases are activated or stabilized in the presence of bile salts, for example bile salt dependent lipase[62,63]. Deoxycholate may act in a similar fashion to stabilize LPL and it might play a role in shifting the lid peptide to an open position, allowing access to the active site[64]. Indeed, we see that the density

corresponding to the lid residues in our map are shifted away from the opening of the active site. This is a different orientation for the lid peptide than was observed when LPL was crystalized in the presence of an active site inhibitor[8]. Given that deoxycholate can serve as a natural activator, this position of the lid, where it has swung to the side, likely reflects the physiological movement triggered by interfacial activation that allows triglycerides to enter the active site. Indeed, there is significant structural overlap between the open lid density in LPL and the open lid peptides in PTL structures (Fig. 3).

LPL had a pronounced preferred orientation in our data, which likely resulted from its interaction with the AWI[38]. We speculate that during grid freezing, LPL interacted with the AWI as though it were a hydrophobic substrate. The interface between the aqueous buffer and air thus mimicked the nonpolar to aqueous interface that triggers interfacial activation of LPL in vivo. Accordingly, the AWI pseudo-substrate favored formation of active dimers of LPL primed for substrate hydrolysis[14,15] with the lid-domain shifted to be in an open state and the active site pore exposed. Our hypothesis is that exposure to a hydrophobic environment is what triggers opening of the pore, as opposed to movement of the lid peptide, which has been shown to be shifted by detergents in other lipase structures[22]. We speculate that

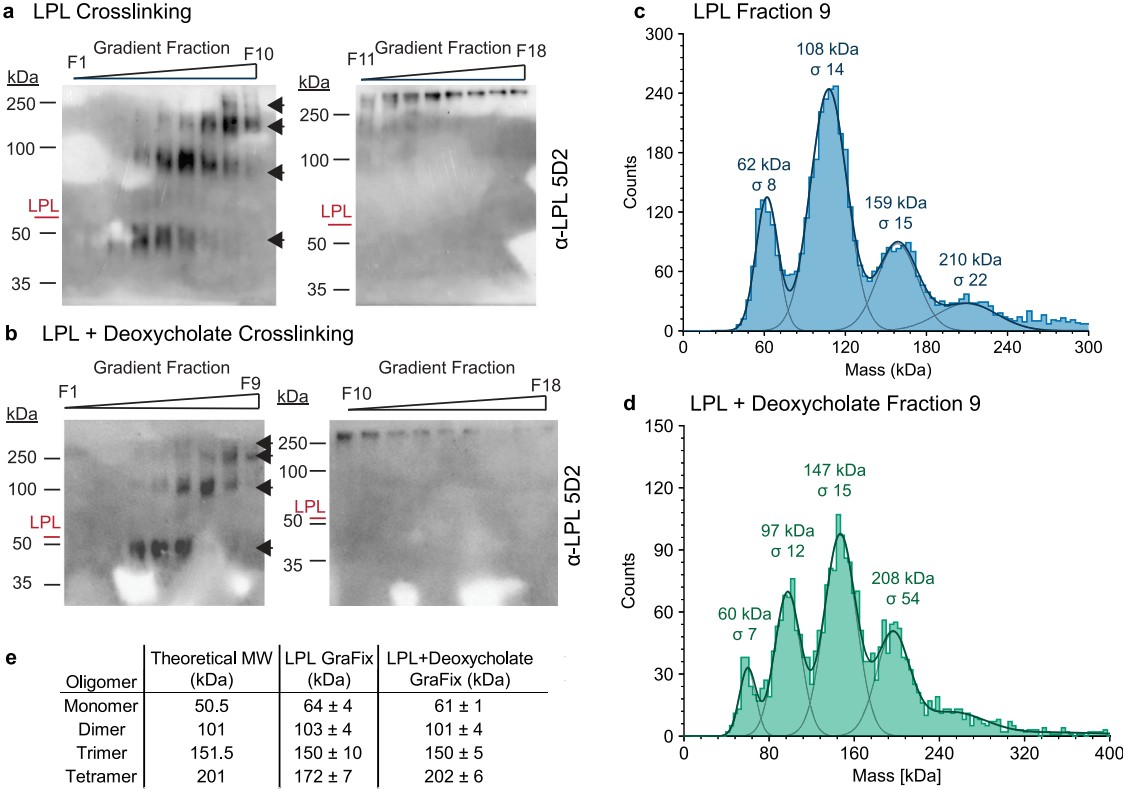

**Fig. 6 | Crosslinking LPL captures monomers, dimers, trimers, and tetramers.** To capture and analyze LPL oligomers, we used gradient fixation (GraFix) to gently crosslink and separate protein by molecular weight. LPL alone (**a**) and with deoxycholate (**b**) showed an array of oligomers forming. The position of uncrosslinked LPL is marked in red. Blots were developed using the monoclonal LPL antibody 5D2, and the first observable LPL oligomer runs below the uncrosslinked LPL molecular weight−likely due to crosslinking altering its electrophoretic mobility. At least 4 distinct LPL oligomers are observed in both samples (black arrows). The similarity of LPL with and without deoxycholate suggests that the deoxycholate does not alter the overall oligomeric state of the protein. GraFix crosslinking experiments were performed in triplicate. The results of the gradient fractionations in (**a**) and (**b**) were analyzed using mass photometry to determine the exact molecular weight of their components. **c** LPL fraction 9 (blue) and **d** LPL with deoxycholate fraction 9 (green) are shown as representative data. The data was fitted with multiple Gaussian distributions to determine the average molecular weight (kDa) and the width of the standard deviation of the fitted Gaussian, sigma (σ). Multiple fractions were analyzed to collect a high quality Gaussian distribution of each oligomer. **e** Results for peak molecular weight Gaussians were averaged and listed in this table as +/− the standard deviation of 3 or more separate measurements. We observe consistent values for monomer, dimer, and trimer for LPL with and without deoxycholate. Molecular weight varied for the likely tetrameric oligomer between the 2 conditions, likely due to fewer data points collected for the LPL tetramer. This difference in distribution can likely be attributed to fractionating by hand and not inherently less tetramer existing in the LPL samples. Source data are provided as a Source Data file.

other lipases may also adopt a preferred orientation at the AWI in cryoEM. The tilting and particle picking strategy described here may be useful for solving structures of other lipases in active states possibly with open hydrophobic pores.

In our cryoEM data we identified classes corresponding to a dimer and tetramer of LPL−but not the monomer or trimer, which we observed only in solution by mass photometry and crosslinking. We believe the AWI pseudo-substrate qualities may explain this discrepancy. It has previously been observed that LPL's oligomeric state is sensitive to the presence of an aqueous-nonpolar interface[5]. In an aqueous solution, such as the one where we performed our crosslinking experiments, LPL was not exposed to a potential substrate interface. This allowed LPL to exist in a variety of oligomeric states, including ones that may be inactive, such as the monomer and trimer. However, during cryo-grid preparation, exposure to the AWI favored the adoption of LPL's active dimer state. As for the tetramer, it would likely be partially active as 2 of the LPL subunits are still oriented to the AWI (Supplementary Figure 8). The other 2 subunits share similarity with the inactive dimer found in the LPL helix, which we suspect may be a result of the high concentration of LPL used to freeze grids. At high concentrations LPL favors helix formation to efficiently pack and store LPL without risk of inactivation[5], but the presence of active LPL dimers due to the AWI and deoxycholate precluded helix formation in this case.

Our structure shows that dimeric LPL exists in vitro, leading to the question of where dimeric LPL might exist in vivo. Originally, when dimeric LPL was identified as the active form of LPL, it was thought that dimeric LPL[18] was attached to HSPGs in the capillaries[65] to interact with lipoproteins. The discovery of the LPL/GPIHBP1 interaction[7], and the fact that LPL forms a heterodimer with it[8,9], changed the conventional thinking on the matter. Previous work has shown significant association of LPL and lipoproteins in post-heparin plasma (heparin dissociates LPL from GPIHBP1)[11,33] and illustrates that LPL does not require GPIHBP1 to associate with a lipoprotein[41]. Nor indeed is GPIHBP1 required for LPL to hydrolyze lipoprotein triglycerides, as free LPL can readily hydrolyze lipoproteins in plasma samples[11,33]. To determine which lipoproteins LPL primarily associates with, plasma samples were treated with a lipase inhibitor immediately following collection, and LPL was found predominantly on VLDL particles[11]. By contrast, when an inhibitor was not used, LPL was found on cholesterol-rich lipoproteins, such as high- and low-density lipoproteins (HDL and LDL)[19]. These data show that LPL remains active on lipoproteins after removal from GPIHBP1 and continues to hydrolyze triglycerides until they are depleted. It has also been shown that in pre-heparin plasma (i.e. untreated plasma) LPL still associates with lipoproteins, specifically VLDL[11]. Recent work using fluorescence to monitor the

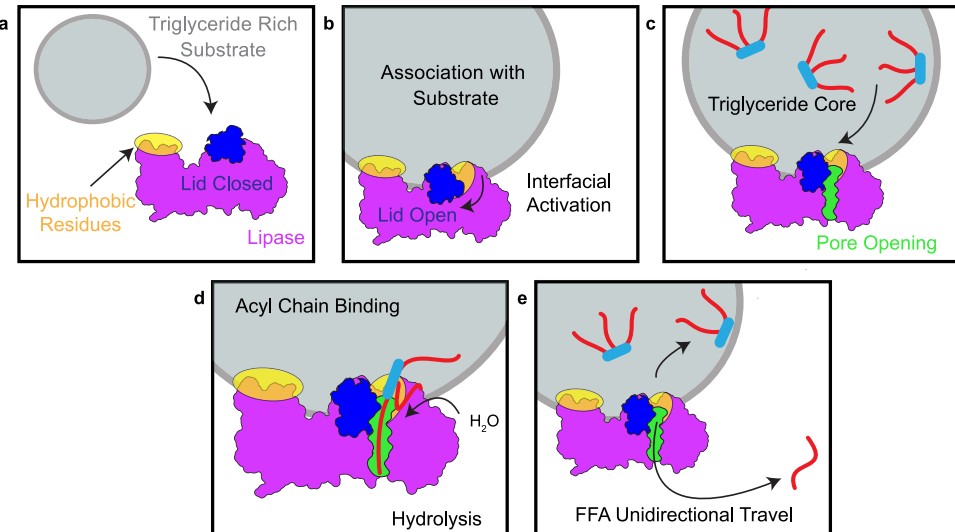

**Fig. 7 | Proposed Unidirectional Model for Triglyceride Hydrolysis by LPL.**
**a** Prior to associating with a substrate (gray), the lid peptide (dark blue) covers access to the LPL (purple) active site. Hydrophobic tryptophan residues (yellow) in the C-terminal domain of LPL are involved in substrate recognition[41]. **b** After association with a substrate, LPL undergoes interfacial activation, where the association with a nonpolar-aqueous interface facilitates the lid peptide swinging to the side. **c** Following lid-peptide opening, the presence of triglyceride (red/light blue) or a hydrophobic environment triggers formation of an acyl-chain binding pore (lime green). **d** The hydrophobic acyl chain (red) enters the pore positioned so that the lipase active site residues can hydrolyze the ester bond connecting the acyl chain to the glyceride backbone (light blue). **e** Following hydrolysis, the free fatty acid (FFA) exits the pore from the opposite side it entered, resulting in unidirectional FFA travel. The remaining ligand can continue to be hydrolyzed at other acyl chains or transfer back to the hydrophobic substrate. This unidirectional model efficiently allows FFA to transfer away from their original source and be bound by proteins in the capillary, such as albumin, that can facilitate their travel to surrounding tissues for continued energy processing.

LPL/GPIHBP1 complex has shown that LPL dissociates from GPIHBP1 in the presence of its substrate, chylomicrons, but also when excess product, free fatty acids (FFA), are added[66]. The release of FFA from chylomicrons by LPL may, therefore, promote dissociation of the LPL/GPIHBP1 complex and formation of the active LPL homodimer[66].

Beyond hydrolyzing triglycerides, LPL is known to associate with lipoprotein remnants and facilitates their uptake by the liver[13]. Prior experiments using sandwich ELISAs showed that dimeric LPL is responsible for mediating lipoprotein binding to the low-density lipoprotein (LDL) receptor-related protein (LRP)[67,68]. Heparin does not affect the association of LPL with lipoproteins but does disrupt LPL interaction with GPIHBP1 and LRP[69]. We hypothesize that following dissociation from GPIHBP1, in the presence of substrate, LPL would favor formation of an active dimer (Fig. 8). This would stabilize LPL and allow it to continuously hydrolyze triglycerides from the core of the lipoprotein until they were depleted. LPL would then remain in its dimeric state to facilitate remnant uptake by the liver[67]. The transition of a chylomicron to a chylomicron remnant entails a reduction in the circumference of the lipoprotein, as triglycerides are hydrolyzed, potentially the resulting alteration in lipoprotein curvature or exposure of new residues from chylomicron constituent proteins could serve as a way to regulate lipoprotein remnant untethering from the capillary wall.

The potential for the unidirectional mechanism for mammalian lipases to facilitate transfer of FFA during hydrolysis provides valuable insight into how lipase mutations can lead to disease. It also opens new arenas of analysis for other mammalian lipases given their significant structural and mechanistic similarity to LPL. It is clear from this work and other recent structures that LPL can adopt a diverse series of oligomeric states. LPL, and lipases in general, play crucial roles in providing energy to the body, but it is equally important that they are active only in the correct milieus. Changes to quaternary structure may allow LPL to self-regulate as it navigates its different roles moving from an inactive form stored in adipocyte vesicles[5], to a heterodimer on the capillary wall[8,9], and finally onto lipoproteins as a homodimer to eventually facilitate uptake by the liver[67].

## Methods

### bLPL purification
Bovine LPL was purified from raw milk using heparin sepharose chromatography[5]. The raw milk was spun at 3565 × g to remove excess fat and the remaining milk was adjusted to 340 mM NaCl with 0.1 mM phenylmethylsulfonyl fluoride (PMSF). The milk solution was incubated with Heparin Sepharose 6 Fast Flow (Cytiva, 17099801) beads for 2 h at 4 °C. The beads were collected and washed with buffers at three different salt concentrations 500 mM, 850 mM, and 950 mM NaCl all with 10 mM Bis-Tris pH 6.5. Protein was eluted at 1.5 M NaCl with 10 mM Bis-Tris pH 6.5. Eluted fractions were dialyzed overnight into in 500 mM NaCl, 10 mM Bis-Tris pH 6.5, and 10% glycerol buffer. The dialyzed protein was loaded onto a HiTrap Heparin HP 5 mL column (Cytiva, 17040701), washed with 500 mM NaCl, 10 mM Bis-Tris pH 6.5, and 10% glycerol, then further washed by a gradient up to 1.1 M NaCl. Protein was eluted in a single step to 2 M NaCl, 10 mM Bis-Tris pH 6.5, and 10% glycerol.

### Lipase activity assay
LPL activity was assayed using an enzyme-coupled fluorescence assay to detect non-esterified fatty acids (NEFA) with VLDL as the substrate[35]. Concentrated LPL (~1 mg/mL) was dialyzed overnight at 4 °C in 500 mM NaCl, 20 mM Tris-HCl pH 8 using a 3.5 kDa slide-a-lyzer. When indicated, 1 mM sodium deoxycholate was added to LPL prior to dialysis. A final concentration of 10 nM LPL was used in assays and the starting concentration of the dialyzed LPL was determined by nanodrop.

Immediately prior to preparing the assay, LPL was diluted to 50 nM in phosphate-buffered saline (PBS) with fetal bovine serum (FBS). The final concentration in the well was 10 nM LPL, 0.2× PBS, and 2% FBS. The reaction was started by the addition of the enzymes needed to detect NEFA, VLDL (Athens Research & Technology,

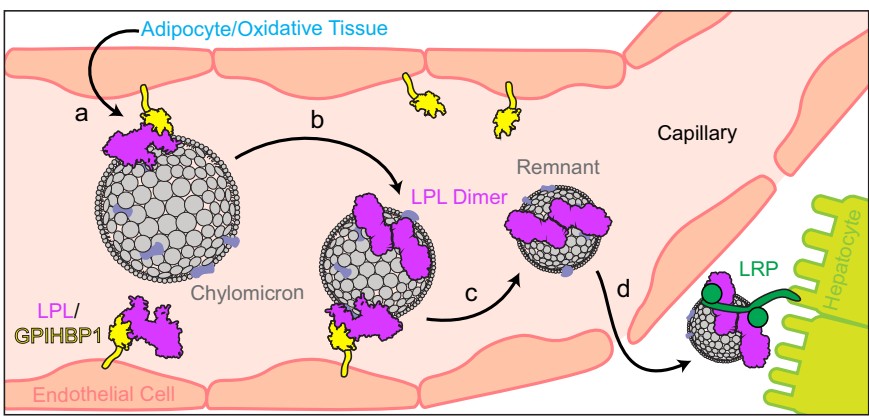

**Fig. 8 | Proposed model of LPL dimer formation in the capillary.** LPL (purple) is synthesized in the adipose and oxidative tissues and secreted into the interstitial space, where it binds to GPIHBP1 (yellow) as a heterodimer[8,9]. **a** GPIHBP1 transports LPL across the endothelium (salmon) into the capillary (pink)[7], where it interacts with triglyceride rich lipoproteins, including chylomicrons (gray) and VLDL[10]. **b** LPL hydrolyzes triglycerides to release free fatty acids which trigger dissociation from GPIHBP1[66]—allowing LPL to form a homodimer, maintaining activity and protein

stability. **c** Following continued triglyceride hydrolysis, the lipoprotein remnant is released from all associations that keep it tethered to the capillary wall and the LPL homodimers remain on the remnant[11]. **d** The lipoprotein remnant and attached LPL homodimer travels through the capillaries to the liver and pass into the space of Disse, where the remnant binds to hepatic transmembrane protein, LRP (green). LRP has previously been shown to interact with LPL homodimers to enhance uptake of low-density lipoprotein remnants into hepatocytes (lime green)[67,68].

12-16-221204), and Amplex UltraRed (ThermoFisher, A36006). The final concentrations in the well from this mixture were: 133 mM KPO₄ pH 7.5, 150 mM NaCl, 3.3 mM MgCl₂, 4.4 mM adenosine triphosphate, 1 mM Coenzyme A (CoA), 0.1 U/mL Acetyl-CoA synthetase (ACS), 6 U/mL horseradish peroxidase, 5 U/mL Acetyl-CoA oxidase (ACO), 0.2 mg/mL fatty-acid free bovine serum albumin (BSA), 0.05 mM Amplex UltraRed, and 200 μg/mL triglycerides in human VLDL. Assays were conducted in a black-walled, 96-well plate. Immediately following addition of the substrate, fluorescence was monitored using a M5 Spectramax plate reader using SoftMax Pro 5 at 37 °C, excitation at 529 nm and emission at 600 nm, with a 590 nm cut-off filter. Initial rate was determined from the first 180 s of the reaction. Three biological replicates were conducted for each assay. Activity assay data was graphed using DataGraph.

### CryoGrid preparation
LPL at a concentration of 1 mg/mL or higher was mixed with 1 mM sodium deoxycholate and dialyzed into 500 mM NaCl, 20 mM Tris-HCl pH 8 at 4 °C either with or without 1 mM deoxycholate in the dialysis buffer. Samples were dialyzed overnight or for a minimum of 5 h. Following dialysis, samples were spun to remove potential aggregates and transferred to a new tube. The concentration of the resulting LPL was verified using a NanoDrop. Several additives were tested to reduce LPL's preferred orientation, but none were successful in altering the preferred orientation. Samples were vitrified using a Vitrobot set at 4 °C and 100% humidity. UltrAuFoil R1.2/1.3 and R0.6/1 300 mesh grids were used. Grids were cleaned using a Tergeo Plasma Cleaner (PieScientific) with settings for direct plasma at 15 W with a 255 duty ratio, in an atmosphere with 2.5% oxygen and 7.5% argon for 1 min. 3 μL of each sample was applied to the grid, pre-blotted for 10 sec, blotted for 3–4 s, and plunge frozen in a liquid ethane/propane mix at −180 °C.

### CryoEM data collection
Grids were imaged using a 200 kV Talos Arctica equipped with a Gatan K3 camera. The microscope was operated using SerialEM[70] and data were collected using beam-image shift with a 5 × 5 grid to speed data collection[71]. Beam-tilt compensation was calibrated in SerialEM to reduce the residual phase error from large beam-image shifts. Data was collected over 2 sessions from two different grids (UltrAuFoil R0.6/1 and R1.2/1.3) prepared with 0.55 mg/mL of LPL that had been dialyzed into 1 mM sodium deoxycholate in 500 mM NaCl and 20 mM Tris-HCl pH 8. All data sets were collected at a nominal magnification of

45,000× (0.88 Å/pixel) with a total flux of 55.1 e⁻/Å². In total 13,748 movies were collected at 0° (4659), 30° (3275), and 45° (5814) tilts.

### CryoEM data processing
All data were processed using the Cryo-EM Single Particle Ab-Initio Reconstruction and Classification (cryoSPARC) software suite[40]. Raw frames were aligned using patch motion correction with dose weighting followed by patch contrast transfer function (CTF) estimation. The data taken for each tilt (0°, 30°, 45°) were kept separate for pre-processing and particle picking. Micrographs with low resolution CTF fits and high total full-frame motion distance were removed leaving 3566 micrographs at 0° tilt, 2135 at 30° tilt, 4525 at 45° tilt.

Initial processing yielded a low-resolution density map, which was highly anisotropic due to a preferred orientation of LPL on the grid. This map was filtered to 15 Å and used to create templates for particle picking and 2D classification. 2D classes with visible secondary structure were selected and filtered for the best 10% of particles per class using the cryoSPARC class probability filter. The remaining 90% of particles from selected classes and the particles from classes that were not selected were combined and subjected to another round of 2D classification. Following multiple rounds of 2D classification, the highest probability particles from the best classes for each round of classification were used to train the Topaz neural network[39] using the cryoSPARC wrapper. Topaz-picked particles were 2D classified and any rare views (non-preferred orientation) were selected, filtered to select the particles that best fit the non-preferred orientation 2D classes, and fed back to Topaz for another round of training. This process was repeated iteratively until no additional rare view particles were found by Topaz picking.

At this point, particles corresponding to rare views were recombined with the preferred orientation particles found in the initial Topaz picking and checked for duplicates. After generating a complete particle set for each tilt using the above method, the unbinned particles for all tilts were extracted using a 320-pixel box size and combined, resulting in a combined dataset of 1,035,101 particles. Ab initio model generation was used to find a subset of particles with reduced anisotropy. This was followed by heterogenous refinement and homogenous refinement with C1 symmetry. We then applied C2 symmetry to the homogenous refinement and took the best resulting model with C2 symmetry, which included 527,205 particles. The result of homogenous refinement was then subjected to non-uniform refinement[72] minimizing over a per-particle scale. A final map with a GSFSC

resolution of 3.9 Å was obtained. Local and global CTF refinement did not improve the map, possibly due to the low molecular weight of the target protein.

The final map was submitted to the 3DFSC server[37], showing a GSFSC resolution of 3.91 Å and a sphericity of 0.85 out of 1. A sphericity of 1 would indicate that the particles that went into making the map sampled every orientation evenly. This indicated that our tilting and picking strategy was able to overcome the preferred orientation of our sample[37]. Analysis of the model found that the hand of the alpha helices was incorrect, so the hand was flipped with cryoSPARC volume tools. The final map was sharpened with a map-sharpening B-factor of −100 Å$^2$. The distribution of particles to the final map was calculated using UCSF PyEM v0.5[73].

### Model building
The structure of an LPL monomer from the inactive LPL helix (PDB 6U7M) was fit into the sharpened map[5] using PHENIX[74,75]. This was followed by iterative rounds of PHENIX real-space refinement and manual adjustment in Coot[76]. C2 symmetry was applied to the model and used to generate the other LPL subunit. One round of PHENIX Rosetta Refine[77] was performed on the LPL dimer, followed by further iterative rounds of PHENIX real-space refinement and adjustment in Coot. The final model has a cross correlation efficient of 0.77 with the map; additional statistics can be seen in Table 1.

### Structure analysis
Structures were visualized using UCSF ChimeraX[78]. To characterize the LPL active site pore, we used MOLEonline[42]. Initial identification of the pore was done with the bottleneck radius parameter set to 1.2 Å with a tolerance of 3. Residues at the top and bottom of the pore were selected as the starting and end points. Overlay of the LPL dimer with other lipase structures was performed in ChimeraX[78].

### Fatty acid substrate modeling
Models for ligands modeled into the LPL pore were obtained from the PDB: palmitate (PLM), oleate (OLA), linoleate (EIC), and triglyceride (TGL). Ligands were fit into the pore manually, overlaying them with the pore representation from MOLEonline[42] in ChimeraX[78]. Coot[76] was used to relax the ligands to reduce clashes. These coot relaxed models were then used to start a PyRosetta[44] minimization run, which allowed both the ligand and LPL to shift in order to generate an improved fit. The PyRosetta modeling resulted in lower repulsion scores for all ligands located in the hydrophobic pore. The models used to start PyRosetta and the final relaxed models are available as Supplementary Data 1–8.

### Mass photometry
Mass photometry was performed with a Refeyn Two MP, which uses the impacts of particles hitting a glass coverslip to determine the molecular weight of the particles—Refeyen Acquire MP and Refeyen Discover MP were used to collect and analyze data respectively[49]. Coverslips were sonicated in isopropanol prior to affixing 3 mm × 1 mm plastic gaskets (Grace Bio-Labs). BSA was diluted into PBS to test the calibration of the mass photometer before each use— confirming a mass of 66 and 133 kDa, for monomer and dimer BSA respectively. BSA was diluted directly into a single gasket containing PBS. The PBS in the gasket prior to addition of protein was used to focus the mass photometer on the glass-buffer interface. Alternatively, new mass calibrations were created with BSA, Urease, and IgM. A stock of 6.25 μM LPL was diluted into PBS or PBS that been pre-mixed with 110 U/mL heparin or 1 mM deoxycholate. LPL was then diluted into a single gasket containing pure PBS at a 1:4 ratio to collect data. The final concentrations of additives were, therefore, 27.5 U/mL heparin and 0.25 mM deoxycholate. Particle impact events were analyzed as ratiometric contrast data and converted to molecular weight using

calibrations with known molecular weight standards[79]. Data for each sample was binned in 3 kDa increments and then analyzed with Origin graphing software to fit single or multiple Gaussian peaks.

Crosslinked samples from GraFix (described below) were similarly diluted directly into a single gasket containing PBS. Concentration of crosslinked samples were below 100 nM for each fraction. We analyzed multiple GraFix fractions to determine the mass of each observed oligomer.

### Gradient fixation (GraFix)
Linear gradients from 5% to 20% glycerol were generated in tubes for an SW-60 rotor. 0.2% glutaraldehyde was added to the 20% glycerol to additionally create a linear glutaraldehyde gradient from 0 to 0.2%. The gradients were cooled to 4 °C and 150 μL LPL with or without deoxycholate was layered on top of the gradient. The gradients were spun at 38,000 RPM (which has an average force of 148,000 × g) for 18 h at 4 °C in an ultracentrifuge. The gradients were removed from the centrifuge and fractionated from the top in 150 μL fractions. The resulting fractions were analyzed by western blot. The protein oligomers were separated by 8% SDS-PAGE, transferred to 0.22 μm PVDF membrane, and blocked with 5% non-fat milk in TBS-T (20 mM Tris-HCl pH 7.6, 150 mM NaCl, and 0.01% Tween 20). LPL was probed with a monoclonal 5D2 LPL antibody (Abcam, ab93898) at a 1:250 dilution and protein was detected with a HRP-conjugated goat anti-mouse antibody (Southern Biotech, 11030-05) using a 1:5000 dilution. Western blots were developed as previously described[80].

Interestingly, another monoclonal LPL antibody, 4-1a (Millipore, MABS1270) at 1:5000 dilution, which binds to the N-terminal fragment of LPL, did not detect crosslinked LPL on these blots, although uncrosslinked controls were detected. To improve our detection resolution, we also performed gradients where we added a final concentration of 1.3 μM TAMRA-FP Serine-Hydrolase probe (TAMRA) (ThermoFisher) to the starting 1 μM LPL mixture. This allowed us to analyze the SDS-PAGE gels without western blotting, using a GE Typhoon scanning at 532 nm. We did not observe any significant differences between the results for western blotting and TAMRA labeling, indicating that the TAMRA probe did not change the oligomeric states adopted by the LPL (Supplementary Fig. 3). The TAMRA labeled samples were used for mass photometry—enabling selection of gradient fractions on the same day as gradient fractionation.

### Human LPL
Human LPL samples were analyzed with mass photometry using the same protocol as bovine LPL. A furin resistant LPL mutant was utilized in these experiments, as larger amounts of uncleaved LPL can be isolated from tissue culture with this mutant[81]. Crosslinking of human LPL (and comparison bovine LPL) was performed in a 500 nM solution of LPL in PBS with 5% glycerol, 5 mM bis(sulfosuccinimidyl)suberate (BS3) (Thermo, A39266), and 1 μM TAMRA for 30 min at RT. Crosslinking results were analyzed by SDS-PAGE using 8% acrylamide gels and then imaged using tamara fluorescence and western blotting as described above.

### Reporting summary
Further information on research design is available in the Nature Portfolio Reporting Summary linked to this article.

## Data availability
The data that support this study are available from public repositories or in the source data file. The Cryo-EM density map was deposited in the Electron Microscopy Data Bank (EMDB) under accession number 28554. Model coordinates have been deposited in the Protein Data Bank (PDB) under accession number 8ERL (LPL homodimer). Other structures used in this study were obtained from the PDB with accession codes 6U7M (LPL helix), PDB 6OB0 (LPL with GPIHBP1), 1LPA

(PTL/procolipase complex), and 1ETH (porcine PTL). Map 20673 (LPL helix) from the EMDB was also used. Models for LPL/ligand complexes are available as Supplementary files 1 through 8, including before and after minimization with PyRosetta. Interface analysis results are available as Supplementary Data 9. Source data are provided with this paper for each figure where relevant raw data exists. Source data are provided with this paper.

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

## Acknowledgements

We thank members of the Neher Lab for discussions and assistance. We thank Dr. Joshua Strauss and Jared Peck at the UNC CryoEM Core and Dr. Nathan Nicely at the UNC Macromolecular Crystallography Core for their assistance. We thank Dr. Brian Kuhlman for assistance with PyRosetta. We thank Dr. Rick Baker and Robert Risti for critical reading of the manuscript. The authors acknowledge support from the National Institutes of Health (R01-HL125654 to S.B.N. and 1K99-GM146024 to K.H.G.) and the American Heart Association (Postdoctoral Fellowship 900354 to K.H.G.).

## Author contributions

K.H.G. and S.B.N. planned the experiments and wrote the manuscript. K.H.G. performed experiments and analyzed data.

## Competing interests

The authors declare no competing interests.
