## [Peer review file · Nature Communications]

REVIEWER COMMENTS

Reviewer #1 (Remarks to the Author):

The manuscript by Gunn and Neher presents a new mechanistic model for the hydrolysis of triglycerides by lipoprotein lipase (LPL) that features a channeling mechanism with the products diacylglycerol and free fatty acid exiting the active site on opposite ends of the enzyme. The work is based on a newly determined structure of bovine LPL that captures LPL in a previously unseen dimeric form that contains a unidirectional pore that is large enough to channel a single fatty acid. The authors hypothesize that the air-water interface induced interfacial activation to capture this conformational state of the lipase. Experiments combining chemical cross-linking and mass photometry support the existence of the observed LPL oligomer state in solution. The work is of high quality and represents an original and important contribution that adds to recent breakthroughs in our knowledge of LPL structure function. There are issues that need to be addressed to fully validate the authors conclusions, which are mentioned below.

Major Comments:

I have several comments and clarifications (points 1-4) based on the mass photometry experiments.

1. As mass photometry is a relatively new technique (at least I was not aware prior), it could be useful to include a sentence in the results section to introduce the technique. This could be something similar to the first sentence in the mass photometry methods section.

2. After reading about the technique in the original describing manuscript (Sonn-Segev et al, 2020: ref 66), it is stated that a current limitation of mass photometry is a protein concentration limit that is required to be less than 100 nM. However, the experiments in Figure 4 that observe mass species of LPL beyond a monomer are either near this limit (78 nM) or over this limit (156 nM). Thus, how do the authors rationalize the use of mass photometry under these experimental conditions to make solid conclusions?

3. Also in Figure 4, the conclusion that LPL is forming a dimer does not seem supported by these experiments (in contrast to Figure 5 with cross-linking) given the observed deconvoluted masses are not close to being multiples of 2. I expected the masses to double in size (e.g. 50 kDa for monomer and 100 kDa for dimer), however the deconvoluted masses are generally much closer to each other (e.g. 59 kDa and 73 kDa; or 95 kDa and 119 kDa). How confident are the authors that this experimental evidence

supports dimerization in solution? Could this be an artifact of using a protein concentration higher than the experimental limit for mass photometry?

4. In figure 5, the cross-linking experiments and mass photometry give much clearer results that support the authors conclusions of dimer formation (and oligomerization beyond) in solution with observed masses that double (dimer), triple (trimer), and quadruple (tetramer); and also match the expected mass sizes. However, it is not stated in the figure legend or methods what protein concentration these experiments were conducted at. This seems to be an important detail given the above mentioned protein concentration limit.

5. In figure 7, one part of the model appears to not be directly supported by experimental evidence. This is the dissociation of LPL from GPIHBP1 to form an LPL dimer. In the discussion section (lines 393-394) the authors mention that “The exact trigger that causes LPL dissociation from GPIHBP1 bears further research.” The authors may consider either providing a reference that demonstrates LPL dissociation from GPIHBP1 or being more explicit about speculation in this part of their model.

6. What species of LPL were the experiments (in references 12, 13, and 15-17) conducted with that demonstrated the existence of an LPL dimer (Lines 88-92)? Human, bovine, or both? Given the observed LPL dimer is from bovine LPL, is their evidence that both bovine and human LPL form homo-dimers? Or just bovine? If all evidence of dimerization comes from bovine LPL, it is certainly worth discussing if LPL from different species may operate by different mechanisms; Alternatively, if evidence is present for bovine, human, and other LPLs it seems pertinent to discuss why LPL from different species most likely share similar regulatory mechanisms and oligomeric states.

7. Since LPL crystal structures are of human LPL, it seems relevant to mention in the initial results experiments that the LPL used for these cryoEM experiments is bovine LPL, and not human LPL for clarity.

8. It would be useful to provide a figure that shows the variation of resolution across the LPL dimer (e.g. as determined by ResMap). In particular, what is the local resolution for the cavity lining residues? In addition, a figure showing the density for the pore lining residues would be improve interpretation for the reader.

9. How was the pore cavity size calculated? Was it calculated from cryoEM density? Or from the protein model?

10. Are there clashes when monoacylglycerol is docked into the unidirectional pore? A downloadable file would be useful for others to analyze.

11. Line 195-196: Which chemical species of monoacylglycerol was modeled in the active site? Is this a palmitate acyl chain (C16:0)? Would an oleate acyl chain (C18:1) with a double bond also fit? Or can the cavity not accommodate a kink induced by a double bond?

Minor comments:

12. Lines 86-87. This statement about other inactive forms beyond a monomer is confusing given the statement directly above “these data suggested the single subunits of LPL (‘monomers’) are the active form ...”

13. Line 163: the statement “associated by the C-termini” is ambiguous given this could refer to the C-terminal domain or the actual C-terminal residue.

14. Line 170-171: suggest adding the word “would” between arrangement and allow

15. Lines 370-384: This paragraph has several statements that are lacking references, which would be useful to include. For example, “originally, when dimeric LPL was identified as the active form of LPL” (add reference(s)) ... and “The discovery of GPIHBP1 ... changed the conventional thinking on the matter” (add reference(s)) ... and “showing that LPL does not require GPIHBP1 to associate with a lipoprotein” (add reference(s))

Reviewer #2 (Remarks to the Author):

An earlier x-ray crystal structure showed an inactive LPL. The hydrophobic C-terminal tryptophan loop, critical for lipid binding, was buried in the hydrophobic catalytic pocket of another LPL molecule. The authors identified, by cryo-EM, a new structure for LPL. In this structure, under the conditions of cryo-EM, the hydrophobic C-terminal region that would normally interact with GPIHBP1’s LU domain interacts with the same hydrophobic C-terminal region in a partner monomer.

They speculate that their cryo-EM structure exists in vivo or in vitro, but there is no evidence. There is no evidence that the multimers observed with glutaraldehyde cross linking are relevant to the cryo-EM structure. These data should be removed.

The fact that LPL can form dimers in vitro at high concentrations has been established earlier. The mass photometry studies provide no evidence for the existence of the conformation that they identified by cryo-EM. It would be possible to address this issue with simple control experiments.

They cherry pick old claims that LPL is dimeric. Most of the data in the older references was marginal in the first place and has been discredited in more recent studies.

In essence, they have uncovered a second conformation for LPL under cryo-EM conditions, but provide no evidence for the existence of this structure in vivo or in vitro.

The cryo-EM model provides for a pore, but there is no evidence that the pore contains a fatty acid, or that fatty acids would traffic unidirectionally through the pore, or that active dimers exist under physiologic conditions, or that a pore exists in monomers (which are highly active). Experimental data is required.

“We hypothesize that the air/water interface formed during creation of samples for cryo-EM triggered interfacial activation, allowing us to capture, for the first time, a fully open state.” Evidence is needed.

Reviewer #3 (Remarks to the Author):

The manuscript by Gunn HK and Neher SB “Structure of dimeric lipoprotein lipase (LPL) reveals a unidirectional pore for hydrolysis of acyl chains” reports the first structure of an active dimer of bovine LPL. The structure was obtained by single particle cryo-EM. Based on analysis of the cryo-EM derived 3D model and literature data, authors postulate, for the first time, existence of a hydrophobic pore next to the active site of LPL, which would play a role in substrate specificity of LPL.

The structure reported in the manuscript is an important development of previous work by the authors, which revealed an inactive oligomeric form of LPL (Gunn et al, 2020 <https://doi.org/10.1073/pnas.1916555117>).

The manuscript is well written and represents a thorough and technically sound work. Reported structure has allowed authors to compare active bovine LPL to other enzymes and accurately pinpoint a dimerization domain and conformational changes leading to opening of the active site. Authors note an unexpected arrangement of the LPL homodimer. However, the main emphasis is made on characterisation of an apparent hydrophobic pore, suggesting that catalysis product movement through it should be conserved among species, but not observed before due to difficulties in studying of LPL in presence of an activating substrate.

While I appreciate technical accuracy of cryo-EM work, I find that the claim of a new mechanism of LPL action based on selectivity of unidirectional pore is not currently supported by sufficient evidence. It is an elegant hypothesis at best at this point but cannot be the central claim of the paper without additional data. Below are the key issues, which I see with such claim in the current manuscript:

1. The “discovery” of the pore is done in-silico using MOLEonline software and is based solely on atomic model fit into cryo-EM density map. However, the reported 3.91 Å resolution of the cryo-EM map makes an accurate fit of many side chains difficult. Author’s comparison of the reported map with higher resolution model of an active human LPL (PDB: 6OB0), shown in Supplemental movie, suggests that some side chains rotated 180° or more to open the pore. This claim, however, is not backed by any figures demonstrating a superior precision of the fit for the side chains modelled into the reported cryo-EM density.
2. No evidence is provided that the pore occupancy by products of catalysis is important for LPL activity or selectivity. Both, the substrate recognition domain and active site are located on the same surface of the molecule. There seems to be no driving force for substrate to enter a narrow channel towards an opposite side of the molecule. If authors insist on validity of their claim, experimental evidence showing consequences of the pore blockage on enzyme activity should be presented.
3. The authors do not provide anisotropic resolution map for the cryo-EM 3D model. This is important, because preferred orientation reported in this manuscript and flexibility reported for other LPL structures can both result in resolution differences for various LPL domains, which in turn is important for validation of atomic model building, especially in the claimed pore region.

Other remarks:

4. For demonstration of the impact of tilted data collection on data completeness, it would be useful to provide in Supplemental materials a map of Euler sphere coverage, indicating a number of particles per 3D model projection in final reconstruction.

5. I find it peculiar, that CTF refinement did not improve the resolution, given the high tilt angle used in data collection. Were results of CTF refinement validated by tracing them back to the micrographs? The refined CTF gradient should be roughly orthogonal to the tilt axis if procedure has worked as expected.

6. Very little attention in the manuscript is given to the exploration of forces stabilising the lid peptide in open conformation to enable substrate access to the active site. Such analysis would be very interesting for understanding biochemistry of enzyme activation.

7. There is a discrepancy between cryo-EM map resolution claimed in the manuscript (3.91 Å) and that reported in Table 1 (4 Å).

Reviewer #4 (Remarks to the Author):

Lipoprotein lipase (LPL) plays a key role in lipid metabolism by hydrolyzing triglycerides present in triglyceride-rich lipoproteins. This manuscript is very noteworthy because the authors have determined for the first time: 1) the structure of an active, dimeric form of LPL, and 2) the presence of a pore or tunnel in the region of the active site of LPL. Neither of these features were seen in previously reported LPL structures. A tunnel was observed in the crystal structure of a *Candida rugosa* lipase (ref 29 in the manuscript) but this manuscript describes the first example of a mammalian lipase structure that features a tunnel. The structure provides novel insights into possible mechanisms of substrate recognition and processing by LPL and suggest that similar mechanisms might be used by other mammalian lipases. The work appears to be very sound technically and is clearly presented. My main suggestion to improve the manuscript prior to publication is to make it more clear which conclusions are based directly on the experimental data and which are hypothesized mechanisms or models that are suggested by the data. Also, some points of discussion currently in the introduction, results and figure legends can be moved to the discussion section.

Specific comments:

1. Title: The data in the paper doesn't directly support a unidirectional mechanism, so I suggest deleting "unidirectional" from the title.
2. Line 41: Adopt a more cautious tone by changing "can" to "may".
3. Line 45: Was the *Candida rugosa* lipase crystal structure also a fully open state of a lipase, with substrate tunnel? If so, consider qualifying this statement to a first observation of a fully open state of LPL.
4. Line 48: The statement "The elucidation of a dimeric LPL structure highlights how LPL can adopt a diverse range of quaternary structure as it travels from secretory vesicles in the cell, to the capillary, and eventually to the liver for lipoprotein remnant uptake" is speculation. Consider deleting this sentence or reword to adopt a more cautious tone.
5. Line 84: Change "the active form" to "an active form" to reflect that GPIHBP1-bound LPL is not the only active form.
6. Line 87: The phrase "also other inactive forms of LPL beyond a monomer" is confusing to me here, since the idea that the monomer is in fact an inactive form has been challenged. Please reword to avoid confusion.
7. Line 116: I suggest qualifying the statement "Given the structural and mechanistic similarities between LPL and other mammalian lipases, the unidirectional mechanism of FFA hydrolysis is likely generalizable". E.g., "proposed unidirectional mechanism" and change "likely generalizable" to "may be utilized by other mammalian lipases".
8. Line 110 and rest of this paragraph. I suggest deferring discussion of possible implications of the pore to the discussion section. I think a general statement here that the structure provides novel insights into potential mechanisms for LPL and possibly other lipases would provide a crisper introduction.
9. Line 138: In what sense was the LPL prep "homogeneous"? How was this shown?
10. Line 165-172: Consider deferring comparison to prior models for LPL dimerization to the discussion section.
11. Line 175: I suggest changing "is engaged" to "could engage".
12. Line 191: Qualify the statement "where the glycerol backbone rests", e.g. to "where we expect the glycerol backbone is located during glyceride hydrolysis".
13. Line 198: Consider deleting the first sentence of this paragraph. As written, it's not clear to me if the sentence refers to the pancreatic lipase structure analysis that follows or if other mammalian lipase structures were evaluated.
14. Line 202: replace "substrates" with "ligands".
15. Line 209-214: "The pore layout we observe suggests that substrate specificity is conferred by a combination of the lid peptide, hydrophobic pocket, and hydrophobic pore. Further, once a triglyceride is hydrolyzed, the diacyl-, monoacyl-glycerol, or glycerol is then able to exit from the entrance of the pore and diffuse back to the surface layer of the chylomicron (41), while the liberated FFA can pass

unidirectionally through the pore and directly into the capillary” can be deferred to the discussion section (and “chylomicron” can be replaced by “TRL”).

16. Line 224: “This suggests that LPL homodimerization and binding to GPIHBP1 are mutually exclusive states. Therefore, when LPL is bound to GPIHBP1, it is not able to form a dimer, but upon dissociation from GPIHBP1, LPL would be able to form a homodimer.” Consider moving this point to the discussion section.

17. Lines 226-230 are confusing – “The homodimerization interface also overlaps with interfaces involved in formation of the inactive LPL helix” is followed by “There is no overlap with the inactive, helical dimerization interface”. These seem to be contradictory statements – please reword to clarify.

18. Line 290: Add references for the statement “LPL has long been known to form an active dimer”.

19. Line 294: The statement “This suggests that binding of the C-terminal domain by an interacting partner is crucial for keeping LPL stable” seems rather speculative to me – consider replacing with the point now in the results section that this indicates LPL dimerization and GPIHBP1 binding are mutually exclusive.

20. Line 298: Mention the C. Rigosa structure early in the paragraph here – the only prior example of a lipase with a tunnel, and clarify as needed the open vs. closed state of the lipase in the C. Rigosa structure.

21. Line 302: I suggest making the point about processivity here – potential to hydrolyze multiple acyl chains without dissociating from the TRL substrate. Change “chylomicron” to “TRL”.

22. Line 355: I suggest qualifying by changing “with open hydrophobic pores” to “possibly with open hydrophobic pores”.

23. Line 398: Qualify by changing “Discovery of” to, e.g. “The potential for”.

24. Line 404: Qualify by changing “allow” to “may allow”.

25. Line 746: The figure caption can be shortened here to avoid redundancy with the results section.

26. Line 815: Consider shortening this figure caption and making these points in the discussion section.

27. Line 822: I suggest making the discussion here more focused on LPL, e.g., association with TRL may trigger pore formation and make the point about potential for generalizing to other mammalian lipases in the text.

Minor comments:

1. Line 48: change “structure” to “structures”.

2. Line 64: Saying the “tissues that surround capillaries” sounds awkward to me since all tissues surround capillaries. Consider rewording.

3. Line 72: modify to reflect that GPIHBP1 is not a transmembrane protein but rather a GPI-anchored protein.

4. Line 140: change "few" to "a few".
5. Line 153: what does "(0.143)" mean?
6. Line 196: change "1-monoacylglycerol" to "C16 1-monoacylglycerol".
7. Line 258: delete "a".
8. Line 286: change "Supplemental Figure B&C" to "Supplemental Figure 5 B&C".
9. Line 385: change "lipoproteins" to "lipoprotein".
10. Line 498: change ",," to ";".

REVIEWER COMMENTS

Reviewer #1 (Remarks to the Author):

The manuscript by Gunn and Neher presents a new mechanistic model for the hydrolysis of triglycerides by lipoprotein lipase (LPL) that features a channeling mechanism with the products diacylglycerol and free fatty acid exiting the active site on opposite ends of the enzyme. The work is based on a newly determined structure of bovine LPL that captures LPL in a previously unseen dimeric form that contains a unidirectional pore that is large enough to channel a single fatty acid. The authors hypothesize that the air-water interface induced interfacial activation to capture this conformational state of the lipase. Experiments combining chemical cross-linking and mass photometry support the existence of the observed LPL oligomer state in solution. The work is of high quality and represents an original and important contribution that adds to recent breakthroughs in our knowledge of LPL structure function. There are issues that need to be addressed to fully validate the authors conclusions, which are mentioned below.

Overall comments: Thank you for your thoughtful analysis of the manuscript. We have worked to address your concerns with regard to mass photometry as detailed below. We have also expanded on our work fitting fatty acids into the pore and added experiments with human LPL to validate our work with bovine LPL.

Major Comments:

I have several comments and clarifications (points 1-4) based on the mass photometry experiments.

1. As mass photometry is a relatively new technique (at least I was not aware prior), it could be useful to include a sentence in the results section to introduce the technique. This could be something similar to the first sentence in the mass photometry methods section.

Thank you for your suggestion, we have added a description.

2. After reading about the technique in the original describing manuscript (Sonn-Segev et al, 2020: ref 66), it is stated that a current limitation of mass photometry is a protein concentration limit that is required to be less than 100 nM. However, the experiments in Figure 4 that observe mass species of LPL beyond a monomer are either near this limit (78 nM) or over this limit (156 nM). Thus, how do the authors rationalize the use of mass photometry under these experimental conditions to make solid conclusions?

Thanks for doing that research. As you stated, this is a pretty new technique. We have looked at other references, and most suggest a 100-200 nM concentration range, which our studies are within (for example, PMID: 35405425, PMID: 31923382). Others suggest less than 1 μ M (PMID: 34608319). That said, the best concentration for mass photometry is often empirical, since the conditions that will give rise to impacts that are numerous enough to provide an interpretable histogram, vary with protein and buffer. However, we repeated experiments with lower concentrations for Figure 3, so they are now all under 100

nM. The data in Figure 5 (LPL concentrations) remain consistent with experiments in Figure 6 (crosslinking). We also added a supplemental figure 6 with human LPL (to address comment 6, below).

3. Also in Figure 4, the conclusion that LPL is forming a dimer does not seem supported by these experiments (in contrast to Figure 5 with cross-linking) given the observed deconvoluted masses are not close to being multiples of 2. I expected the masses to double in size (e.g. 50 kDa for monomer and 100 kDa for dimer), however the deconvoluted masses are generally much closer to each other (e.g. 59 kDa and 73 kDa; or 95 kDa and 119 kDa). How confident are the authors that this experimental evidence supports dimerization in solution? Could this be an artifact of using a protein concentration higher than the experimental limit for mass photometry?

In this figure, we are fitting multiple gaussians to molecules likely in rapid equilibrium between oligomeric states. Note the more expected molecular weight fits in the cross-linked molecules, which aren't in equilibrium. On the whole, we want to illustrate for the reader that the peaks corresponding to the average mass shifts as the concentration of LPL increases. Mass photometry is useful here because it is solution phase measurement. LPL is a very sticky protein that causes problems with techniques using a lot of surface area such as chromatography. We thus believe that mass photometry is the best way to get a solution phase look at LPL's oligomeric diversity and its concentration dependence. We prefer to leave the figure in, although we have tried to adjust how we present the data in the text and made the changes mentioned above.

4. In figure 5, the cross-linking experiments and mass photometry give much clearer results that support the authors conclusions of dimer formation (and oligomerization beyond) in solution with observed masses that double (dimer), triple (trimer), and quadruple (tetramer); and also match the expected mass sizes. However, it is not stated in the figure legend or methods what protein concentration these experiments were conducted at. This seems to be an important detail given the above mentioned protein concentration limit.

Thanks for your observation. We have added this detail, and the protein is under the limit.

5. In figure 7, one part of the model appears to not be directly supported by experimental evidence. This is the dissociation of LPL from GPIHBP1 to form an LPL dimer. In the discussion section (lines 393-394) the authors mention that "The exact trigger that causes LPL dissociation from GPIHBP1 bears further research." The authors may consider either providing a reference that demonstrates LPL dissociation from GPIHBP1 or being more explicit about speculation in this part of their model.

We have edited the text to reference the following publication: A novel NanoBiT-based assay monitors the interaction between lipoprotein lipase and GPIHBP1 in real time. PMID: 32029511. In this publication, the authors can monitor binding of LPL to GPIHBP1 located on endothelial cells in real time. Importantly, the authors discovered that chylomicrons and FFA are able to induce LPL dissociation from GPIHBP1. We have also been more explicit about our speculation involving the triggers of LPL dissociation from GPIHBP1.

6. What species of LPL were the experiments (in references 12, 13, and 15-17) conducted with that demonstrated the existence of an LPL dimer (Lines 88-92)? Human, bovine, or both? Given the observed LPL dimer is from bovine LPL, is their evidence that both bovine and human LPL form homo-dimers? Or just bovine? If all evidence of dimerization comes from bovine LPL, it is certainly worth discussing if LPL from different species may operate by different mechanisms; Alternatively, if evidence is present for bovine, human, and other LPLs it seems pertinent to discuss why LPL from different species most likely share similar regulatory mechanisms and oligomeric states.

All of our experiments, except the new supplemental figure 6, are carried out with bovine LPL. We have made this more explicit in the text. Bovine and human LPL have 94% identity, and 97% similarity. Based on this conservation, it is very unlikely that there are any major differences in the properties of these two proteins. We did, however, add supplemental figure 6, which replicates the mass photometry data with lower concentrations and crosslinking data with human LPL.

7. Since LPL crystal structures are of human LPL, it seems relevant to mention in the initial results experiments that the LPL used for these cryoEM experiments is bovine LPL, and not human LPL for clarity.

Thank you, we have clarified this.

8. It would be useful to provide a figure that shows the variation of resolution across the LPL dimer (e.g. as determined by ResMap). In particular, what is the local resolution for the cavity lining residues? In addition, a figure showing the density for the pore lining residues would be improve interpretation for the reader.

Thank you for your suggestion. The figure has been added (Supplemental Figure 3).

9. How was the pore cavity size calculated? Was it calculated from cryoEM density? Or from the protein model?

The pore cavity was calculated from the protein model. We've clarified this in the text.

10. Are there clashes when monoacylglycerol is docked into the unidirectional pore? A downloadable file would be useful for others to analyze.

We have now included the pdb files of the fatty acids fit into the LPL pore as supplemental files, including starting fits and fits minimized with PyRosetta. We also included a model for a triglyceride in the pore (Supplemental Figure 5). The PyRosetta output includes a value for the repulsion (fa_rep) of the ligand and all residues following minimization, so we can be confident that we have limited clashes with the PyRosetta models compared to the starting models. For the starting models we minimized clashes by using Coot, although we did not allow the LPL residues to adjust, so some clashes remained and can be evaluated with the pdb files.

11. Line 195-196: Which chemical species of monoacylglycerol was modeled in the active site? Is this a palmitate acyl chain (C16:0)? Would an oleate acyl chain (C18:1) with a double bond also fit? Or can the cavity not accommodate a kink induced by a double bond?

Indeed, the original Figure 2 had a monoacylglycerol with a palmitate acyl chain, we have now clarified the type of acyl chain fit in the pore in the figure legend. We went ahead and edited Figure 2 to also model in an oleate (C18:1) and linoleate (C18:2) in addition to the palmitate (C16:0) to show that the cavity can theoretically accommodate all of these acyl chains. And as mentioned, we have provided pdb files in the supplement of these models.

Minor comments:

12. Lines 86-87. This statement about other inactive forms beyond a monomer is confusing given the statement directly above “these data suggested the single subunits of LPL (‘monomers’) are the active form ...”

We have rewritten this statement for clarity.

13. Line 163: the statement “associated by the C-termini” is ambiguous given this could refer to the C-terminal domain or the actual C-terminal residue.

Thank you, we clarified this.

14. Line 170-171: suggest adding the word “would” between arrangement and allow.

Thank you, we changed this.

15. Lines 370-384: This paragraph has several statements that are lacking references, which would be useful to include. For example, “originally, when dimeric LPL was identified as the active form of LPL” (add reference(s)) ... and “The discovery of GPIHBP1 ... changed the conventional thinking on the matter” (add reference(s)) ... and “showing that LPL does not require GPIHBP1 to associate with a lipoprotein” (add reference(s))

We added these references.

Reviewer #2 (Remarks to the Author):

Overall comments: We acknowledge your comments and have responded to them below.

An earlier x-ray crystal structure showed an inactive LPL. The hydrophobic C-terminal tryptophan loop, critical for lipid binding, was buried in the hydrophobic catalytic pocket of another LPL molecule. The authors identified, by cryo-EM, a new structure for LPL. In this structure, under the conditions of cryo-EM, the hydrophobic C-terminal region that would

normally interact with GPIHBP1's LU domain interacts with the same hydrophobic C-terminal region in a partner monomer.

They speculate that their cryo-EM structure exists *in vivo* or *in vitro*, but there is no evidence. There is no evidence that the multimers observed with glutaraldehyde cross linking are relevant to the cryo-EM structure. These data should be removed.

The multimers were identified from experiments performed *in vitro* – cryoEM is closely associated with solution based-structures, since the protein is rapidly frozen in buffer, which is akin to the solution state of the protein. In this instance, we hypothesize that the air-water interface of the cryo-grids has played a role in creating a different result for our cryoEM and solution data. Therefore, we believe that the data on oligomeric state from solution is both interesting and useful for thinking about our structure and the oligomeric diversity of LPL in general. We have also validated the glutaraldehyde crosslinking with bis(sulfosuccinimidyl)suberate crosslinking (Supplemental Figure 6).

The fact that LPL can form dimers *in vitro* at high concentrations has been established earlier. The mass photometry studies provide no evidence for the existence of the conformation that they identified by cryo-EM. It would be possible to address this issue with simple control experiments.

The mass photometry data shows that LPL dimers do form in solution, and this is not impacted by the addition of deoxycholate. Therefore, these experiments are relevant to evaluation of our cryoEM structure. This new technique applied to LPL also gives us novel insight into the concentration dependence of oligomer formation. We have now included the mass photometry data as supplemental files and clarified the controls we used in the methods section.

They cherry pick old claims that LPL is dimeric. Most of the data in the older references was marginal in the first place and has been discredited in more recent studies.

As you noted previously, “the fact that LPL can form dimers *in vitro* at high concentrations” has been previously established. Our manuscript includes multiple and references to studies that have used different techniques to demonstrate the existence of an LPL dimer (including studies published in 2018, 2004, 2002, 1993 and 1983). It is also worth noting that many of these references are shared with the pioneering LPL structure paper from 2019 by Birrane et al. (PMID 30559189). The precedent set by this 2019 work, shows that that it is appropriate to reference previous literature showing that an LPL dimer exists. An extract from this paper states:

“For many years, LPL has been assumed to form a head-to-tail homodimer (19, 48–50), and recent studies have proposed that homodimer formation is required for LPL secretion from cells (51).”

While the dimerization of the LPL in this 2019 structure was shown by later work to be a result of the crystallization process (hence why it was not cited as evidence of an LPL dimer), it shows that in the recent past discussions of LPL oligomers are in line with what we have written. We also believe that the multiple oligomeric forms of LPL that have been observed are all bona fide structural forms that exist in different conditions.

In essence, they have uncovered a second conformation for LPL under cryo-EM conditions, but provide no evidence for the existence of this structure *in vivo* or *in vitro*.

CryoEM is an *in vitro* condition. CryoEM conditions are arguably similar to solution conditions, with the difference of vitreous ice and the air-water interface. Beyond cryoEM, our crosslinking and mass photometry are also *in vitro* studies that show evidence of a dimeric LPL oligomer. Work that would show the *in vivo* evidence of dimers is beyond the scope of this study. There is also extensive previous work (as mentioned above) that has shown dimers exist in contexts that are closer to *in vivo*, such as on lipoproteins (PMID: 8978491).

The cryo-EM model provides for a pore, but there is no evidence that the pore contains a fatty acid, or that fatty acids would traffic unidirectionally through the pore, or that active dimers exist under physiologic conditions, or that a pore exists in monomers (which are highly active). Experimental data is required.

The model of how the LPL pore may bind to an acyl chain or triglyceride is a model. We are confident that later studies will be undertaken to address the possible contents of this pore, but they are beyond the scope of this investigation. We have changed the language to emphasize that what we are proposing is a model. The ability of a tunnel from *Candida rugosa* lipase (PDB 1LPP) to bind to acyl chains however does suggest that it is feasible for a pore, such as the one in our structure, to accommodate a fatty acid. We have also shown that we can successfully model in a variety of free fatty acids into the pore (Supplemental figure 5, supplemental files).

“We hypothesize that the air/water interface formed during creation of samples for cryo-EM triggered interfacial activation, allowing us to capture, for the first time, a fully open state.” Evidence is needed.

This is a hypothesis, as stated above and we have emphasized this in the manuscript. We came to this hypothesis by considering the difference between cryoEM and crystallography techniques. The obvious evidence for the air-water interface mimicking a substrate is that the LPL dimer has undergone interfacial activation (as can be seen by comparing to other lipases that have undergone interfacial activation in Figure 3). This is not a process that is known to occur spontaneously, and it unlikely that this activation would be replicated across the hundreds of thousands of individual proteins analyzed in our study without some external impetus.

Reviewer #3 (Remarks to the Author):

The manuscript by Gunn HK and Neher SB “Structure of dimeric lipoprotein lipase (LPL) reveals a unidirectional pore for hydrolysis of acyl chains” reports the first structure of an active dimer of bovine LPL. The structure was obtained by single particle cryo-EM. Based on analysis of the cryo-EM derived 3D model and literature data, authors postulate, for the first time, existence of a hydrophobic pore next to the active site of LPL, which would play a role in substrate specificity of LPL.

The structure reported in the manuscript is an important development of previous work by the authors, which revealed an inactive oligomeric form of LPL (Gunn et al, 2020 <https://doi.org/10.1073/pnas.1916555117>).

The manuscript is well written and represents a thorough and technically sound work. Reported structure has allowed authors to compare active bovine LPL to other enzymes and accurately pinpoint a dimerization domain and conformational changes leading to opening of the active site. Authors note an unexpected arrangement of the LPL homodimer. However, the main emphasis is made on characterization of an apparent hydrophobic pore, suggesting that catalysis product movement through it should be conserved among species, but not observed before due to difficulties in studying of LPL in presence of an activating substrate.

While I appreciate technical accuracy of cryo-EM work, I find that the claim of a new mechanism of LPL action based on selectivity of unidirectional pore is not currently supported by sufficient evidence. It is an elegant hypothesis at best at this point but cannot be the central claim of the paper without additional data. Below are the key issues, which I see with such claim in the current manuscript:

Overall response: The lack of density in the pore region is a prominent feature of the structure. It extends exactly from the active site and is the size of the product of the reaction. This feature is very different from the 3 other LPL structures that have been solved. Indeed, it is unique in mammalian lipase structures, making it important to discuss the pore, but we will amend our language to be clear that the unidirectional nature of the pore is speculative and will require further research – we have also edited the title to remove the word unidirectional. Proving the unidirectionality of this pore is beyond the scope of this work because it would require years of work to solve another structure and extensive collaboration with a chemist to trap this state. However, we have hopefully been able to better explain previous research that shows mutations in the pore result in long-chain substrate deficiency to support our hypothesis. We appreciate your concerns and have added figures, text and modeling to address them as detailed below.

1. The “discovery” of the pore is done in-silico using MOLEonline software and is based solely on atomic model fit into cryo-EM density map. However, the reported 3.91 Å resolution of the cryo-EM map makes an accurate fit of many side chains difficult. Author’s comparison of the reported map with higher resolution model of an active human LPL (PDB: 6OB0), shown in Supplemental movie, suggests that some side chains rotated 180° or more to open the pore. This claim, however, is not backed by any figures demonstrating a superior precision of the fit for the side chains modelled into the reported cryo-EM density.

Thanks for your observation. We will clarify in the text that we don't know the precise way that the residues move to make the pore. The morph is only shown to demonstrate the differences between the two structures. We now include a morph between another cryoEM structure determined at similar resolution (inactive, 6U7M) (Supplemental movie 2), and a morph between PDB: 6OB0 and 6U7M (Supplemental movie 3). The former has a lot of movement, just like the original morph, and the later has minimal movement in the pore region (the lid is quite mobile though). Thus, this suggests that the change we see is not just a matter of resolution or technique, but rather that the two structures are different. We have also added supplemental figure 4 to better illustrate the fitting of the residues lining the pore.

2. No evidence is provided that the pore occupancy by products of catalysis is important for LPL activity or selectivity. Both, the substrate recognition domain and active site are located on the same surface of the molecule. There seems to be no driving force for substrate to enter a narrow channel towards an opposite side of the molecule. If authors insist on validity of their claim, experimental evidence showing consequences of the pore blockage on enzyme activity should be presented.

In the literature we find that prior experimental evidence for the importance of the pore, although this was not understood as such at the time that research was published. We expand on this in the paper.

Briefly, two mutations known to have deleterious effects on human LPL activity localize to the pore exit. The first, a L252R mutation, substitutes a hydrophobic leucine residue with a bulky, positively charged arginine residue. Residue 225 is positioned directly inside of the pore close to the exit. Molecular modelling suggests that this bulkier arginine residue will block the pore (New supplemental Figure 10). *In vitro* studies revealed that arginine or proline substitutions at LPL residue 252 resulted in production of 30% or 70% of WT protein, respectively (Ma et al., JLR, 1994). However, both LPL variants had no specific activity with a long chain triglyceride substrate, specifically a triolein emulsion. Thus, although mutant LPL was produced, this enzyme could not hydrolyze long chain triglycerides in a substrate emulsion.

The second mutation, R75S, results in loss of positive charge at the pore exit (Figure 2 and Supplemental Figure 10). LPL with the R75S mutation was expressed in and purified from cultured cells, with a reduced amount of active enzyme produced relative to the WT protein (Wilson et al, 1993). Interestingly, when corrected for protein concentration, the WT LPL showed higher specific activity on a long chain, triolein substrate relative to LPL R75S, but LPL R75S showed higher specific activity on the small soluble substrate, para-nitrophenyl butyrate.

Thus, both LPL pore mutants identified in chylomicronemic patients could fold and be secreted. However, both variants showed a specific defect on long chain triglyceride substrates, suggesting the pore observed in our structure has physiological relevance for hydrolysis of specifically the long chain substrates, which would fit in the hydrophobic pore.

We would also like to apologize for the confusion of how these residues are numbered – prior to the first structure of LPL numbering was done after removal of the signal sequence. We have explained this in the paper in an effort to limit confusion and used the numbering that matches the existing structures of LPL rather than the older manuscripts.

With regards to what might drive the substrate into the pore, the structure of *Candida rugosa* lipase in complex with substrate mimics, shows that hydrophobic acyl chains can indeed enter a tunnel of comparable size to the LPL pore (1LPP). Therefore, it is not out of line to speculate the pore in our structure could also house an acyl chain. Further research will be required to solve a structure of LPL in complex with a substrate mimic.

3. The authors do not provide anisotropic resolution map for the cryo-EM 3D model. This is important, because preferred orientation reported in this manuscript and flexibility reported for other LPL structures can both result in resolution differences for various LPL domains, which in turn is important for validation of atomic model building, especially in the claimed pore region.

Thank you for your suggestion. This map is now provided as supplemental figure 3. The pore region, which is located in the interior of the protein, actually has higher resolution than the overall resolution of 3.9 Å.

Other remarks:

4. For demonstration of the impact of tilted data collection on data completeness, it would be useful to provide in Supplemental materials a map of Euler sphere coverage, indicating a number of particles per 3D model projection in final reconstruction.

Thanks for your suggestion, we have added this in Supplemental Figure 3.

5. I find it peculiar, that CTF refinement did not improve the resolution, given the high tilt angle used in data collection. Were results of CTF refinement validated by tracing them back to the micrographs? The refined CTF gradient should be roughly orthogonal to the tilt axis if procedure has worked as expected.

While we cannot definitively say why the CTF refinement did not benefit our structure, it could be due in part to the small size of our protein (100 kDa). Our understanding is that small proteins generally see less resolution improvement with CTF refinement. It is also possible that the use of patch CTF refinement at the start of the data analysis process, limited the benefits of downstream CTF refinement, as we were already using CTF estimates of smaller areas than the entire micrograph.

6. Very little attention in the manuscript is given to the exploration of forces stabilising the lid peptide in open conformation to enable substrate access to the active site. Such analysis would be very interesting for understanding biochemistry of enzyme activation.

The lid is unfortunately very mobile, in all structures resolved to date, excepting 60BO, which was solved with an active site inhibitor, which stabilized the lid residues in a likely

unnatural position. We can't provide a thorough analysis about the forces stabilizing the lid peptide in an open position without a high-resolution density to fit residues into. We have clarified, though, in our discussion to suggest that the air-water interface itself may be what is enabling us to capture the lid open state, since it potentially triggered LPL to undergo interfacial activation.

7. There is a discrepancy between cryo-EM map resolution claimed in the manuscript (3.91 Å) and that reported in Table 1 (4 Å).

Thank you for catching this! We have fixed it.

Reviewer #4 (Remarks to the Author):

Lipoprotein lipase (LPL) plays a key role in lipid metabolism by hydrolyzing triglycerides present in triglyceride-rich lipoproteins. This manuscript is very noteworthy because the authors have determined for the first time: 1) the structure of an active, dimeric form of LPL, and 2) the presence of a pore or tunnel in the region of the active site of LPL. Neither of these features were seen in previously reported LPL structures. A tunnel was observed in the crystal structure of a *Candida rugosa* lipase (ref 29 in the manuscript) but this manuscript describes the first example of a mammalian lipase structure that features a tunnel. The structure provides novel insights into possible mechanisms of substrate recognition and processing by LPL and suggest that similar mechanisms might be used by other mammalian lipases. The work appears to be very sound technically and is clearly presented. My main suggestion to improve the manuscript prior to publication is to make it more clear which conclusions are based directly on the experimental data and which are hypothesized mechanisms or models that are suggested by the data. Also, some points of discussion currently in the introduction, results and figure legends can be moved to the discussion section.

Overall comments: Thank you for your thorough analysis of the manuscript. We have taken all of these comments and incorporated them into the text. If figures were added, we noted it below.

Specific comments:

1. Title: The data in the paper doesn't directly support a unidirectional mechanism, so I suggest deleting "unidirectional" from the title. **Done.**
2. Line 41: Adopt a more cautious tone by changing "can" to "may". **Done.**
3. Line 45: Was the *Candida rugosa* lipase crystal structure also a fully open state of a lipase, with substrate tunnel? If so, consider qualifying this statement to a first observation of a fully open state of LPL.

We've clarified by saying the first mammalian lipase. We're not able to definitively say whether the candida rugosa lipase structure is fully open.

4. Line 48: The statement “The elucidation of a dimeric LPL structure highlights how LPL can adopt a diverse range of quaternary structure as it travels from secretory vesicles in the cell, to the capillary, and eventually to the liver for lipoprotein remnant uptake” is speculation. Consider deleting this sentence or reword to adopt a more cautious tone.

We have edited this section to make clear what has been shown versus speculation/hypothesis.

5. Line 84: Change “the active form” to “an active form” to reflect that GPIHBP1-bound LPL is not the only active form.

We have edited this section to clarify.

6. Line 87: The phrase “also other inactive forms of LPL beyond a monomer” is confusing to me here, since the idea that the monomer is in fact an inactive form has been challenged. Please reword to avoid confusion.

We have edited this section to clarify.

7. Line 116: I suggest qualifying the statement “Given the structural and mechanistic similarities between LPL and other mammalian lipases, the unidirectional mechanism of FFA hydrolysis is likely generalizable”. E.g., “proposed unidirectional mechanism” and change “likely generalizable” to “may be utilized by other mammalian lipases”. **Done.**

8. Line 110 and rest of this paragraph. I suggest deferring discussion of possible implications of the pore to the discussion section. I think a general statement here that the structure provides novel insights into potential mechanisms for LPL and possibly other lipases would provide a crisper introduction. **Done.**

9. Line 138: In what sense was the LPL prep “homogeneous”? How was this shown?

We have added a gel of the protein preparation in supplemental figure 1 to illustrate the purity of the LPL used for these experiments.

10. Line 165-172: Consider deferring comparison to prior models for LPL dimerization to the discussion section. **Done.**

11. Line 175: I suggest changing “is engaged” to “could engage”. **Done.**

12. Line 191: Qualify the statement “where the glycerol backbone rests”, e.g. to “where we expect the glycerol backbone is located during glyceride hydrolysis”. **Done.**

13. Line 198: Consider deleting the first sentence of this paragraph. As written, it’s not clear to me if the sentence refers to the pancreatic lipase structure analysis that follows or if other mammalian lipase structures were evaluated. **Done.**

14. Line 202: replace “substrates” with “ligands”. **Done.**

15. Line 209-214: “The pore layout we observe suggests that substrate specificity is conferred by a combination of the lid peptide, hydrophobic pocket, and hydrophobic pore. Further, once a triglyceride is hydrolyzed, the diacyl-, monoacyl-glycerol, or glycerol is then able to exit from the entrance of the pore and diffuse back to the surface layer of the chylomicron (41), while the liberated FFA can pass unidirectionally through the pore and directly into the capillary” can be deferred to the discussion section (and “chylomicron” can be replaced by “TRL”). **Done.**

16. Line 224: “This suggests that LPL homodimerization and binding to GPIHBP1 are mutually exclusive states. Therefore, when LPL is bound to GPIHBP1, it is not able to form a dimer, but upon dissociation from GPIHBP1, LPL would be able to form a homodimer.” Consider moving this point to the discussion section. **Done.**

17. Lines 226-230 are confusing – “The homodimerization interface also overlaps with interfaces involved in formation of the inactive LPL helix” is followed by “There is no overlap with the inactive, helical dimerization interface”. These seem to be contradictory statements – please reword to clarify.

We have edited this section to clarify our meaning. Only some (not all) of the interfaces seen in the LPL helix overlap with the homodimer interface. We also shifted a supplemental figure into the main text (now Figure 4) to make this distinction easier to visualize for the reader.

18. Line 290: Add references for the statement “LPL has long been known to form an active dimer”. **Done.**

19. Line 294: The statement “This suggests that binding of the C-terminal domain by an interacting partner is crucial for keeping LPL stable” seems rather speculative to me – consider replacing with the point now in the results section that this indicates LPL dimerization and GPIHBP1 binding are mutually exclusive. **Done.**

20. Line 298: Mention the C. Rigosa structure early in the paragraph here – the only prior example of a lipase with a tunnel, and clarify as needed the open vs. closed state of the lipase in the C. Rigosa structure. **Done.**

21. Line 302: I suggest making the point about processivity here – potential to hydrolyze multiple acyl chains without dissociating from the TRL substrate. Change “chylomicron” to “TRL”. **Done.**

22. Line 355: I suggest qualifying by changing “with open hydrophobic pores” to “possibly with open hydrophobic pores”. **Done.**

23. Line 398: Qualify by changing “Discovery of” to, e.g. “The potential for”. **Done.**

24. Line 404: Qualify by changing “allow” to “may allow”. **Done.**

25. Line 746: The figure caption can be shortened here to avoid redundancy with the results section. **Done.**

26. Line 815: Consider shortening this figure caption and making these points in the discussion section. **Done.**

27. Line 822: I suggest making the discussion here more focused on LPL, e.g., association with TRL may trigger pore formation and make the point about potential for generalizing to other mammalian lipases in the text. **Done.**

Minor comments:

1. Line 48: change “structure” to “structures”. **We prefer it worded as it currently is.**

2. Line 64: Saying the “tissues that surround capillaries” sounds awkward to me since all tissues surround capillaries. Consider rewording. **Done.**

3. Line 72: modify to reflect that GPIHBP1 is not a transmembrane protein but rather a GPI-anchored protein. **Thank you for catching this! We have corrected it.**

4. Line 140: change “few” to “a few”. **We prefer it worded as it currently is.**

5. Line 153: what does “(0.143)” mean? **We have edited the manuscript to reflect that this is the cut-off for the directional FSC used to determine the reported resolution of our structure.**

6. Line 196: change “1-monoacylglycerol” to “C16 1-monoacylglycerol”. **The carbon number of the ligands are now laid out in the text of Figure 2.**

7. Line 258: delete “a”. **Done.**

8. Line 286: change “Supplemental Figure B&C” to “Supplemental Figure 5 B&C”. **Done.**

9. Line 385: change “lipoproteins” to “lipoprotein”. **Done.**

10. Line 498: change “,” to “;”. **Done.**

REVIEWERS' COMMENTS

Reviewer #1 (Remarks to the Author):

All my comments have been addressed.

Reviewer #3 (Remarks to the Author):

The manuscript by K. H. Gunn and S. Neher “Structure of Dimeric Lipoprotein Lipase Reveals a Pore for Hydrolysis of Acyl Chains” has significantly improved after the first revision. The work is performed accordingly to the accepted standards in the field. Based on their structural studies, the authors propose a novel mechanism for the lipoprotein lipase activation and enzyme processivity.

However, I would like to raise following issues:

Authors have modified the title to remove word “unidirectional”. Yet, I still believe that there is an overstatement in the title. In the current form it implies that studies have been performed demonstrating hydrolysis of acyl chains inside the pore. This is not the case in the current study. Obviously, it is up to authors to name their work, but based on the science described so far in their manuscript, an appropriate title would sound like “Structure of Dimeric Lipoprotein Lipase suggests an existence of a hydrophobic pore capable to accommodate Acyl Chains”. Therefore, either a proof of the acyl chain hydrolysis in the pore has to be provided, or the title should be toned down. Some of the experiments which could help, are suggested below.

In order to back the manuscript claim about the functionality of the pore, I would see it only logical that authors would perform a cryo-EM study of the molecule in the presence of the ligand(s), for example those fitted into the pore in the Figure 2D. If their hypothesis is correct, even at low resolution one can produce a difference map demonstrating the pore occupancy versus an empty structure reported so far.

In the answer to my remark [6] in the initial review, authors claim that the mobility of the lid prevents resolving it. However, 3D classification methods are long established in the cryo-EM to enable extracting useful information for the mobile regions (for review see [doi: 10.1016/j.sbi.2017.07.007]). The size of the data set over 1M particles enables application of such methods. More novel methods include multibody refinement [doi: 10.1007/978-1-0716-0966-8_7], or neural network based methods (see for example [doi: 10.1038/s41592-020-01049-4]).

Chapter “Comparison of active LPL to other open lipase structure”, Line 207-209: “The hydrophobic pore is unique to our structure, which we hypothesize could be due to LPL’s response to the AWI during cryo-grid preparation.” Does this mean that the claimed pore is simply an artefact of sample preparation? I do not see such statement as logical, since the pore is buried inside the structure, while AWI affects surface layers. Deoxycholate treatment could have a much bigger impact on the structure in and around the pore. Authors hypothesis that AWI interactions influence the lid domain (either by trapping it in an open conformation or by disordering it, and hence opening the active site) is much more logical. I would remove or change this sentence.

Minor remark:

In the chapter “Active site pore in the LPL dimer structure” Line 170-171 “suggesting an active conformation that could engaged with a TRL substrate...” should it be “suggesting an active conformation that could be engaged with a TRL substrate...”

Reviewer #4 (Remarks to the Author):

The authors' revised manuscript effectively addresses the main concern I raised, which was to clarify what conclusions are directly supported by the experimental data in the paper, and which are speculation/models/hypotheses inspired by the results in the paper.

In sum, I support publication of the revised manuscript. A few minor suggestions for finalization of the manuscript prior to publication:

Line 91: suggest deleting "beyond a monomer" from "other inactive forms of LPL beyond a monomer" since the start of this paragraph states that the inactive monomer hypothesis has been challenged recently.

Line 111: suggest deleting "those with" and change "implications" to "possible implications" to maintain a more cautious tone.

Line 208: consider rewording "unique to our structure" or adding "among mammalian lipases" or "among LPL structures" to account for the prior publication of a non-mammalian lipase with a pore.

Line 328: add "in which" before "the tunnel".

Line 378: this paragraph can be the end of the previous paragraph.

Line 332: suggest changing "mammalian lipases" to "LPL". The point about potential generalizability is in my opinion better made later on, where you have it on line 351 and again in the conclusion.

REVIEWERS' COMMENTS

Reviewer #1 (Remarks to the Author):

All my comments have been addressed.

Thank you for your review, we're glad the revisions addressed your concerns.

Reviewer #3 (Remarks to the Author):

The manuscript by K. H. Gunn and S. Neher "Structure of Dimeric Lipoprotein Lipase Reveals a Pore for Hydrolysis of Acyl Chains" has significantly improved after the first revision. The work is performed accordingly to the accepted standards in the field. Based on their structural studies, the authors propose a novel mechanism for the lipoprotein lipase activation and enzyme processivity.

Thank you for your review, we've responded to the points you raised below.

However, I would like to raise following issues:

Authors have modified the title to remove word "unidirectional". Yet, I still believe that there is an overstatement in the title. In the current form it implies that studies have been performed demonstrating hydrolysis of acyl chains inside the pore. This is not the case in the current study. Obviously, it is up to authors to name their work, but based on the science described so far in their manuscript, an appropriate title would sound like "Structure of Dimeric Lipoprotein Lipase suggests an existence of a hydrophobic pore capable to accommodate Acyl Chains". Therefore, either a proof of the acyl chain hydrolysis in the pore has to be provided, or the title should be toned down. Some of the experiments which could help, are suggested below.

To address your concern with the title we have changed it to: Structure of Dimeric Lipoprotein Lipase Reveals a Pore Adjacent to the Active Site.

In order to back the manuscript claim about the functionality of the pore, I would see it only logical that authors would perform a cryo-EM study of the molecule in the presence of the ligand(s), for example those fitted into the pore in the Figure 2D. If their hypothesis is correct, even at low resolution one can produce a difference map demonstrating the pore occupancy versus an empty structure reported so far.

We agree that the next extension of our work is to solve the structure of LPL with a substrate in the hydrophobic pore. There are a couple of challenges to overcome for this to be successful:

- 1. The hydrophobicity of triglycerides makes them difficult to solubilize, which would be necessary for looking at single particles (i.e. we need single LPL + single substrate rather than LPL on an emulsified lipid droplet).**
- 2. LPL processes triglycerides very quickly, trapping or crosslinking the substrate in the pore will be needed to create a homogenous sample.**

3. We haven't found any commercially available substrates that are non-hydrolyzable or crosslinkable and balance hydrophobicity with solubility.
4. We will need to collaborate with a chemist in order to develop such a substrate and that would both take extensive time and warrant a separate publication.

With regards to the idea that we could use the free fatty acids we computationally fit into the pore, those acyl chains are the product of the hydrolysis reaction. LPL undergoes product inhibition, so mixing product with the protein would not be a viable option for trapping an active enzyme.

In the answer to my remark [6] in the initial review, authors claim that the mobility of the lid prevents resolving it. However, 3D classification methods are long established in the cryo-EM to enable extracting useful information for the mobile regions (for review see [doi: 10.1016/j.sbi.2017.07.007]). The size of the data set over 1M particles enables application of such methods. More novel methods include multibody refinement [doi: 10.1007/978-1-0716-0966-8_7], or neural network based methods (see for example [doi: 10.1038/s41592-020-01049-4]).

Unfortunately, we have not had success using multibody refinement on our structure. This is a common issue for small proteins that has been observed by many labs, including those who specialize in pushing resolution limits for sub 100 kDa proteins, like Gabe Lander (doi: 10.1042/BST20210360). Essentially, multibody refinement is successful when there is sufficient signal in the area being refined to align the particle. A smaller mass gives less signal, making small sections of proteins, such as the lid domain of LPL (1.5 kDa), unresolvable by the current limits of this technique. Indeed, masked sections under 100 kDa are generally not currently amenable to multibody refinement.

Chapter "Comparison of active LPL to other open lipase structure", Line 207-209: "The hydrophobic pore is unique to our structure, which we hypothesize could be due to LPL's response to the AWI during cryo-grid preparation." Does this mean that the claimed pore is simply an artefact of sample preparation? I do not see such statement as logical, since the pore is buried inside the structure, while AWI affects surface layers. Deoxycholate treatment could have a much bigger impact on the structure in and around the pore. Authors hypothesis that AWI interactions influence the lid domain (either by trapping it in an open conformation or by disordering it, and hence opening the active site) is much more logical. I would remove or change this sentence.

We have moved this hypothesis to the discussion section, where we have the space to discuss it more thoroughly. The AWI is essentially a transition from a hydrophilic buffer environment to a nonpolar air environment. Our theory is that the AWI acted as a large substrate mimic, i.e. mimicking the phospholipid monolayer interface surrounding a triglyceride rich lipoprotein. We know that exposure to a hydrophilic/nonpolar interface triggers movement of the lid peptide, as can detergents like deoxycholate. Crystal structures of lipases have been captured with the lid in an open position in the presence of detergents, but a hydrophobic pore has not been observed. This certainly suggests that it is possible to separate the requirements for opening of the lid peptide and the opening of the hydrophobic pore.

Minor remark:

In the chapter “Active site pore in the LPL dimer structure” Line 170-171 “suggesting an active conformation that could engaged with a TRL substrate...” should it be “suggesting an active conformation that could be engaged with a TRL substrate...”

Thank you for catching that typo! It has been fixed.

Reviewer #4 (Remarks to the Author):

The authors' revised manuscript effectively addresses the main concern I raised, which was to clarify what conclusions are directly supported by the experimental data in the paper, and which are speculation/models/hypotheses inspired by the results in the paper.

In sum, I support publication of the revised manuscript. A few minor suggestions for finalization of the manuscript prior to publication:

Thank you for your review – we have made all of your suggested edits.

Line 91: suggest deleting "beyond a monomer" from "other inactive forms of LPL beyond a monomer" since the start of this paragraph states that the inactive monomer hypothesis has been challenged recently.

We removed the wording as suggested.

Line 111: suggest deleting "those with" and change "implications" to "possible implications" to maintain a more cautious tone.

This paragraph has been edited to provide a more complete summary of the work removing this sentence.

Line 208: consider rewording "unique to our structure" or adding "among mammalian lipases" or "among LPL structures" to account for the prior publication of a non-mammalian lipase with a pore.

Edited to reflect it is unique among mammalian lipases.

Line 328: add "in which" before "the tunnel".

We added ‘in which’.

Line 378: this paragraph can be the end of the previous paragraph.

We combined these two paragraphs.

Line 332: suggest changing "mammalian lipases" to "LPL". The point about potential generalizability is in my opinion better made later on, where you have it on line 351 and again in the conclusion.

We changed to 'LPL'.